# RFMAMBA: FREQUENCY-AWARE STATE SPACE MODEL FOR RF-BASED HUMAN-CENTRIC PERCEPTION

**Rui Zhang[1], Ruixu Geng[1], Yadong Li[2], Ruiyuan Song[1], Hanqin Gong[1],**
**Dongheng Zhang[1], Yang Hu[1], Yan Chen[1]***
[1]University of Science and Technology of China,  [2]University of Washington
`{ruizh,gengruixu,rysong,hanqin_gong}@mail.ustc.edu.cn,`
`yadongli@uw.edu, {dongheng,eeyhu,eecyan}@ustc.edu.cn`

## ABSTRACT

Human-centric perception with radio frequency (RF) signals has recently entered a new era of end-to-end processing with Transformers. Considering the long-sequence nature of RF signals, the State Space Model (SSM) has emerged as a superior alternative due to its effective long-sequence modeling and linear complexity. However, integrating SSM into RF-based sensing presents unique challenges including the fundamentally different signal representation, distinct frequency responses in different scenarios, and incomplete capture caused by specular reflection. To address this, we carefully devise a dual-branch SSM block that is characterized by adaptively grasping the most informative frequency cues and the assistant spatial information to fully explore the human representations from radar echoes. Based on these two branches, we further introduce an SSM-based network for handling various downstream human perception tasks, named RFMamba. Extensive experimental results demonstrate the superior performance of our proposed RFMamba across all three downstream tasks. To the best of our knowledge, RFMamba is the first attempt to introduce SSM into RF-based human-centric perception.

## 1 INTRODUCTION

Despite significant advancements in vision-based human perception (Cao et al., 2023), optical cameras face fundamental limitations when perceiving human motion in non-line-of-sight (NLoS) or low-lighting scenarios (Geng et al., 2021; 2022). In contrast, radio frequency (RF) signals are illumination-robust and can penetrate through non-metallic materials such as concrete walls (Li et al., 2024a;b; Song et al., 2022a; Ding et al., 2023; Wang et al., 2021). Existing studies have utilized various RF signals for human sensing, including Stepped-Frequency Continuous Wave (SFCW) radar, Frequency-Modulated Continuous Wave (FMCW) radar, and WiFi. Among these, SFCW and FMCW signals are well-suited for fine-grained human sensing because their larger bandwidth. Leveraging the signals reflecting off human body, we can track suspects hidden inside a building or quickly locate victims trapped within collapsed structures, making it vital for anti-terrorism and rescue operations.

Prior works on RF-based human-centric perception (HCP) predominantly follow two paradigms. The first approach follows a multi-stage pipeline, where RF heatmaps are first generated from radar signals and subsequently fed into neural networks to perform human perception. This multi-stage process, however, requires complex data preprocessing that results in reduced efficiency. The second approach employs a single-stage framework, exemplified by the RadarFormer model proposed by (Zheng et al., 2023b). They utilize a Transformer-based architecture (Vaswani et al., 2017) to directly extract human representations from radar echoes, thereby eliminating the need for intermediate heatmap generation. This end-to-end process enables more efficient handling of various HCP tasks.

However, due to the long-sequence nature of radar echoes Geng et al. (2024), a single-stage framework that adopts a Transformer-based architecture suffers from quadratic computational overhead, resulting in prohibitively high computational complexity. Therefore, a natural question arises: *can a more efficient yet effective solution be developed to capture the long-range dependencies across large-sequence RF?* The recently popular State Space Model (SSM) with linear complexity could be a

---

*Corresponding author.

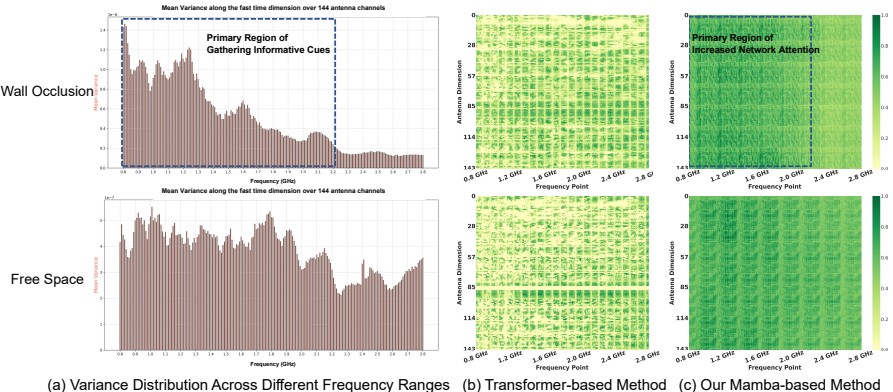

Figure 1: (a) illustrates the variance response across different frequency in the human motion scenarios, where higher variance indicates more captured human information. (b) and (c) illustrate the effective receptive field (ERF) of the transformer-based model and our SSM-based model, respectively. The ERF demonstrates the model's response to diverse frequency ranges. Notably, only the proposed RFMamba can adaptively select the most informative frequency cues for different scenarios.

promising answer to this question. Nevertheless, due to the inherent physical characteristics of RF signals, three key challenges arise when adapting the SSM architecture for RF-based sensing:

**Signal Representation:** The representation of RF signals is fundamentally different from images or text. RF signals consist of amplitude and phase information, with amplitude denoting target reflectivity while phase enables precise distance and velocity estimation.

**Frequency Response:** The signals in different frequency bands have varying abilities to penetrate obstacles, resulting in different frequency responses between the free-space scenario and the wall-occlusion scenario. As shown in Figure 1, the human motion information tends to concentrate in all frequency bands in free space, while concentrating in low-frequency bands under wall occlusion.

**Specular Reflection:** The human body acts as a reflector rather than a scatterer in the lower RF frequency range, causing a single RF echo can only capture partial human limbs. Such specular reflections render the problem of RF-based human perception inherently ill-posed.

Expanding upon these signal characteristics, we first attempt to extend Mamba from the perspective of frequency analysis and propose a frequency-aware Mamba for RF-based HCP, termed RFMamba. Specifically, we devise a RF-State Space Model (RF-SSM) block with two core branches modeling in the frequency domain and spatio-temporal domain, respectively. Due to the three-dimension nature of RF, the frequency modeling branch correlates frequency cues in the amplitude and phase dimensions, and a six-way (omni-dimensional) scanning strategy is adopted to ensure each element in the echoes integrates information from all other locations in different dimensions. Additionally, a frequency adaptive feed-forward network is designed to adaptively identify the most informative frequency cues. Aside from the frequency branch, we also develop a spatio-temporal modeling branch to better facilitate the learning process from the perspective of the spatial domain. Finally, the frequency information and the spatial assistant information are comprehensively interacted to enhance RF-based human perception. With an elaborate model design, our model can handle variable input sequence lengths, relying on multiple echoes to alleviate the problem of specular reflection when using a single echo. RFMamba outperforms state-of-the-art methods by locating human key points with an average error of 5.06 cm, identifying person ID and action category with a mean average precision of 0.9991 and F1 score of 0.9994, respectively.

Our main contribution can be summarized as:

- We pioneer the first state space model for RF-based human perception, demonstrating the potential of Mamba for efficient yet effective global modeling in long-sequence RF signals.
- We introduce a novel RF-SSM block which integrates both frequency domain and spatio-temporal domain modeling to effectively capture critical characteristics of RF signals for human perception.
- We propose a six-way scanning strategy in the frequency modeling branch, which ensures comprehensive interaction of amplitude and phase information across all dimensions and is capable of adaptively selecting the most informative frequency cues.

## 2  RELATED WORK

**Radio Frequency Sensing.** Traditional RF sensing methods primarily rely on advanced signal processing techniques to develop explicit models that link signal variations to human behaviors. However, as the complexity of sensing tasks increases, these models often fall short. Consequently, learning-based RF sensing, driven by machine learning techniques, has emerged as a promising alternative (Zhao et al., 2018a; Xie et al., 2023; Gong et al., 2024). Recent advancements have explored various RF modalities: WiFi-based methods like GoPose (Ren et al., 2022) and Person-in-WiFi 3D (Yan et al., 2024) achieve accurate 3D pose estimation using commodity devices. MmWave approaches such as mmPose-NLP (Sengupta & Cao, 2023) and HuPR (Lee et al., 2023) leverage point clouds for precise pose reconstruction. Expanding on these, milliFlow (Ding et al., 2025) enhances motion sensing with scene flow estimation, mmDiff (Fan et al., 2025) uses diffusion models to improve pose estimation in noisy environments, and RadarOcc (Ding et al., 2024) processes 4D radar data to deliver robust 3D occupancy predictions. The MM-Fi dataset (Yang et al., 2023) further accelerates progress in RF sensing by offering a multi-modal benchmark for various perception tasks. UWB systems like UWB-Pose (Song et al., 2022b) and MD-Pose (Zhou et al., 2023) exploit wide bandwidth for through-wall applications. While these methods have progressed, they often rely on complex preprocessing (Wu et al., 2022) or struggle with long sequence efficiency (Zheng et al., 2023b). Recent RadarFormer (Zheng et al., 2023b) introduced end-to-end processing of radar echoes but only utilized amplitude information. Our work addresses these limitations by introducing a novel state space model, efficiently processing both amplitude and phase information in long RF sequences.

**State Space Models.** State Space Models (SSMs) have emerged as a powerful alternative to traditional architectures like Transformers (Peng et al., 2024; Han et al., 2024; Ye et al., 2025; Jin et al., 2025), particularly for tasks involving long-range dependencies due to their linear complexity with input length (Gu et al., 2020; 2021). Initial work such as S4 (Gu et al., 2022) and S5 (Smith et al., 2023) laid the foundation for deep state-space models, demonstrating their ability to model sequences efficiently. This was followed by the introduction of Mamba (Gu & Dao, 2024), which enhanced SSMs by incorporating a selective scan mechanism and efficient hardware design, making it highly competitive in various domains. Recent works have adapted Mamba for diverse tasks, including vision (Zhu et al., 2024), medical imaging (Ma et al., 2024; Liu et al., 2024a), and video understanding (Chen et al., 2024). However, these adaptations primarily focus on single-modality data or specific domain applications. RFMamba distinguishes itself as the first to optimize SSMs for RF-based human-centric perception. Unlike existing Mamba-based models, RFMamba addresses the unique challenges of RF signals through novel components like the RF-SSM block and omni-dimensional scanning strategy. This approach enables efficient processing of both amplitude and phase information in long RF sequences, a crucial aspect not addressed by Mamba models in other domains.

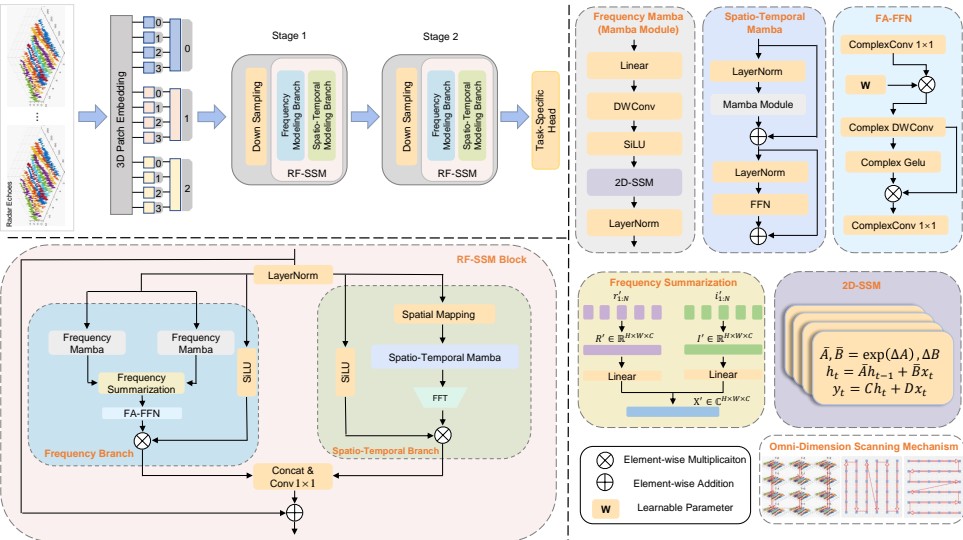

Figure 2: The overview of our state space model for RF-based HCP. RFMamba consists of a 3D patch embedding module, a two-stage encoder, and a task-specific head.

## 3  METHOD

### 3.1  PRELIMINARY

#### 3.1.1  FREQUENCY AND SPATIAL ANALYSIS IN RF SENSING

In RF-based human sensing, Stepped-Frequency Continuous Wave (SFCW) signals are widely used due to their high range resolution and penetration capabilities. The SFCW signal model can be expressed as:

$$s(t) = \sum_{n=0}^{N-1} a_n e^{j2\pi f_n t},$$  (1)

where $N$ is the number of frequency steps, $a_n$ is the amplitude, and $f_n = f_0 + n\Delta f$ is the frequency at the $n$-th step, with $f_0$ being the start frequency and $\Delta f$ the frequency step size.

The received RF signal can be formulated as $X = \mathcal{R} + j \cdot \mathcal{I} \in \mathbb{C}^{A \times F}$, where $A$ represents the number of antennas and $F$ the number of frequency samples. The corresponding amplitude spectrogram $\mathcal{A}$ and phase spectrogram $\mathcal{P}$ can be extracted by:

$$\mathcal{A}(u,v) = \sqrt{\mathcal{R}^2(u,v) + \mathcal{I}^2(u,v)}, \quad \mathcal{P}(u,v) = \arctan[\frac{\mathcal{I}(u,v)}{\mathcal{R}(u,v)}],$$  (2)

where $u$ and $v$ indicate the indices in the frequency domain. The inverse Fourier transform converts $X$ to the spatial domain $S(h,w)$:

$$S(h,w) = \frac{1}{\sqrt{AF}} \sum_{u=0}^{A-1} \sum_{v=0}^{F-1} X(u,v) e^{j2\pi \left( \frac{h}{A}u + \frac{w}{F}v \right)}.$$  (3)

This transformation allows analysis in three critical dimensions: angle ($h$), range ($w$), and velocity (from range variations across frames). Our RFMamba leverages this multi-dimensional information, incorporating both frequency and spatial domain features through the proposed RF-SSM block and omni-dimensional scanning strategy, enabling comprehensive modeling of human motion information.

#### 3.1.2  STATE SPACE MODELS

As a novel basic operation, State Space Models (SSMs) capture long-term dependencies similar to self-attention but benefit from linear complexity, which shows efficient scalability with input length. By leveraging the content of linear ordinary differential equations to translate one-dimensional inputs into outputs via latent states, SSMs operate with high efficiency. For a system with input $x(t)$ and output $y(t)$, SSMs can be formulated as a linear ordinary differential equation:

$$h'(t) = \mathbf{A}h(t) + \mathbf{B}x(t), \quad y(t) = \mathbf{C}h(t) + \mathbf{D}x(t),$$  (4)

where $h(t)$ is the hidden state that accumulates historical information, and $h'(t)$ describes its temporal evolution. $\mathbf{A} \in \mathbb{R}^{N \times N}$ governs state transitions, $\mathbf{B} \in \mathbb{R}^N$ maps inputs to state updates, $\mathbf{C} \in \mathbb{R}^N$ projects states to outputs, and $\mathbf{D} \in \mathbb{R}^1$ provides direct input-output connections. For the through-wall radar system, the SFCW signals are processed using zero order hold (ZOH) discretization, enabling adaptive scanning of RF data for human motion reconstruction. This adaptability is particularly useful in RF sensing to reconstruct human motion from long-sequence discrete RF signals.

### 3.2  OVERALL ARCHITECTURE

As is illustrated in Figure 2, RFMamba is a multi-stage network consisting of a 3D patch embedding module, a two-stage encoder, and a task-specific head. Specifically, to alleviate the problem of fragmented target information in a single frame due to specular reflection, we input $T$ sequential radar echoes with the shape $\mathbf{F}_e \in \mathbb{C}^{(T \times A \times F)}$ into the patch embedding module, which can preserve the structural and sequential relationships of echoes. Then, a two-stage encoder is employed to obtain the high-dimensional representation. Each stage consists of a down-sampling layer and an RF-SSM block. The down-sampling operation is implemented using a complex convolution layer

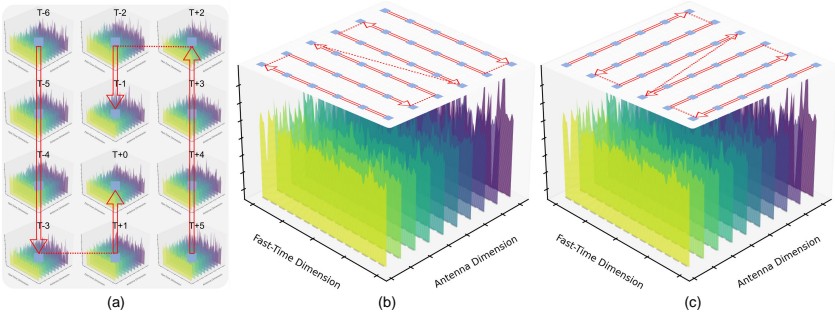

Figure 3: The proposed omni-dimensional scanning strategy. In the frequency domain, (a), (b), and (c) represent scanning methods along the slow-time, fast-time, and antenna dimensions, respectively. In the spatial domain, (a) corresponds to scanning along the temporal dimension, while (b) and (c) represent scanning along different spatial dimensions.

with a kernel size of $1 \times 3 \times 3$ and a stride of $1 \times 2 \times 2$, resulting in a 2x down-sampling of the resolution $\frac{F}{2} \times \frac{A}{2}$. The RF-SSM block includes the frequency modeling branch which captures long-term dependencies from amplitude and phase domain, and the spatiotemporal modeling branch which extracts distance, angle, and velocity features. The outputs from these branches are integrated into a unified high-dimensional representation, which is processed by a lightweight task-specific head to generate predictions for downstream tasks.

### 3.3 3D PATCH EMBEDDING MODULE

To better utilize the temporal (slow-time dimension) information, we merge the 12 sequential radar echoes as the layout in Figure 3 (a), which can realize a fixed scanning interval along the temporal dimension. After the merge operation, the shape of $\mathbf{F}_e$ becomes $\mathbf{F}_e \in \mathbb{C}^{(A \cdot T_1) \times (F \cdot T_2)}$. We adopt 2D complex convolution to project the input $\mathbf{F}_e$ into $N$ non-overlapping 2D patches $\mathbf{F}_p \in \mathbb{C}^{(H \times W \times N)}$, where $H = 32$ and $W = 39$. Note that the 2D patches are not further flattened into a 1D sequence, which can preserve the 2D structure of the radar echoes. Furthermore, to preserve the information regarding the spatial and temporal positions, we further incorporate a learnable spatial position embedding, denoted as $\mathbf{p}_s \in \mathbb{C}^{1 \times A \times F}$, alongside an additional temporal position embedding, represented as $\mathbf{p}_t \in \mathbb{C}^{T \times 1 \times 1}$. Then, $\mathbf{p}_s$ and $\mathbf{p}_t$ are expanded into the same shape $T \times A \times F$ by broadcast operation. The patches $\mathbf{F}_p$ are initially rearranged into the shape at $T \times A \times F$, after adding the learnable spatial and temporal position embedding, the patches $\mathbf{F}_p$ are rearranged into the shape at $H \times W \times N$. This procedure is encapsulated as:

$$\mathbf{F}_p = \mathbf{F}_p + \mathbf{p}_s + \mathbf{p}_t. \tag{5}$$

### 3.4 RF-STATE SPACE MODEL BLOCK

The detailed structure of RF-SSM block is shown in Figure 2, which contains two parallel frequency and spatiotemporal branches to extract frequency and spatiotemporal information. The frequency branch captures long-term dependency from the amplitude and phase of RF signals, while the spatiotemporal branch captures the spatio-temporal information (range, angle, and velocity) from the RF signal with an additional FFT operation. The frequency information and the spatiotemporal information interact with each other to generate the high-level semantic representations, significantly improving the performance of RF-based human-centric perception.

### 3.5 FREQUENCY MODELING BRANCH

#### 3.5.1 OMNI-DIMENSIONAL SCANNING MECHANISM

Existing Mamba architectures(Zhu et al., 2024) highly depend on scanning directions, such as scanning natural language from left to right and scanning natural images from four directions. The radar signals are 3-dimension data, consisting of the fast-time dimension with distance information, the antenna dimension with azimuth and elevation information, and the slow-time dimension with

velocity information. To scale the Mamba for radar signals, we propose a scanning strategy that encapsulates comprehensive omni-dimensional information while maintaining linear computational complexity. Specifically, we combined 12 consecutive radar echoes as shown in Figure 3. The scanning strategy along the slow-time dimension is illustrated in Figure 3 (a), and the scanning strategy along the fast-time dimension and antenna dimension is shown in Figure 3 (b). The omni-dimensional scanning mechanism includes six scanning directions: forward antenna dimension, forward fast-time dimension, forward slow-time dimension, backward antenna dimension, backward fast-time dimension, and backward slow-time dimension. Subsequently, the features from six scanning directions are merged to form the final features, allowing the model to effectively capture global frequency information.

### 3.5.2 FREQUENCY MAMBA MODULE

The feature map $\mathbf{F}_p$ in the frequency domain consists of two parts, i.e., amplitude spectrogram $\mathcal{A}(\mathbf{F}_p)$ and the phase spectrogram $\mathcal{P}(\mathbf{F}_p)$. These two spectrograms are processed individually using the progressive frequency scanning branch depicted in Figure 2 to obtain $\mathcal{A}'(\mathbf{F}_p)$ and $\mathcal{P}'(\mathbf{F}_p)$.

$$\mathcal{A}'(\mathbf{F}_p) = \mathrm{FreqScan}(\mathcal{A}(\mathbf{F}_p)), \quad \mathcal{P}'(\mathbf{F}_p) = \mathrm{FreqScan}(\mathcal{P}(\mathbf{F}_p)). \tag{6}$$

where $\mathrm{FreqScan}$ indicates the frequency scanning process. The detailed of $\mathrm{FreqScan}$ is shown in Figure 3. The $\mathrm{FreqScan}$ contains a series of operations: $Linear \rightarrow DWConv \rightarrow SiLU \rightarrow SS2D \rightarrow LayerNorm \rightarrow Linear$. Unlike the vanilla scanning strategy (Liu et al., 2024b), our scanning strategy is the omni-dimensional frequency learning. The amplitude and phase scanning information are further integrated by frequency summarization module as

$$\mathbf{F}_p = Linear(A'(\mathbf{F}_p)) + j \cdot Linear(P'(\mathbf{F}_p))). \tag{7}$$

### 3.5.3 FA-FFN MODULE

Not all low- and high-frequency information contributes to latent human perception. To address this, we propose a Frequency Adaptive Feed-Forward Network (FA-FFN) module, which adaptively identifies the relevant frequency components to focus on. The FA-FFN can be formulated as follows:

$$X_3 = \mathrm{DWConv}(\mathrm{Conv}_{1 \times 1}(\mathbf{F}_p) \odot W_{fft}), \quad \mathbf{F}_p = \mathrm{GELU}(X_3) \cdot X_3. \tag{8}$$

The input feature map $\mathbf{F}_p$ is first passed through a $1 \times 1$ complex convolution layer, which projects it into a higher-dimensional space.

The transformed features are element-wise multiplied by a learnable weight matrix $W_{fft}$, followed by complex depthwise convolution (DWConv) to generate $X_3$. Subsequently, the GELU activation function is applied to introduce non-linearity, and the result is further multiplied by $X_3$, producing the final output $\mathbf{F}_p$.

### 3.6 SPATIOTEMPORAL MODELING BRANCH

The heatmaps in the spatial domain are considered to contain latent information related to distance, angle, and velocity. In light of this, we design a spatiotemporal modeling branch, which assists the learning process from a spatial domain perspective. The detailed structure of the spatiotemporal branch is shown in Figure 2.

### 3.6.1 SPATIAL MAPPING MODULE

By converting radar echoes from the frequency domain into heatmaps in the spatial domain, the spatial mapping module, which utilizes both IFFT and FFT, enables detailed local analysis. The radar echoes captured by the radar system are denoted as $\mathbf{F}_e \in \mathbb{C}^{T \times F \times A_1 \times A_2}$, where $A_1 \times A_2 = A$. First, a 1D IFFT is applied along the fast-time dimension $F$ to extract the target's distance information. Then, a 1D FFT is applied along the antenna dimensions $A_1$ and $A_2$ to capture angle information of azimuth and elevation. The spatial mapping module performs these transformations along the fast-time dimension, transmitting antenna array, and receiving antenna array, respectively. The absolute value function is then applied to obtain the spatial heatmap $\mathbf{F}_s$, formulated as:

$$\mathbf{F}_s = \mathrm{ABS}(\left(F_F^{-1}(\mathbf{F}_e), F_{AT}(\mathbf{F}_e), F_{AR}(\mathbf{F}_e)\right)). \tag{9}$$

By stacking $T$ spatial heatmaps $\mathbf{F}_s$, the spatiotemporal heatmap $\mathbf{F}_{st}$, which contains both spatial and temporal information, is obtained. The arrangement of multi spatial heatmaps can be seen in Figure 3. The $\mathbf{F}_{st}$ is then fed into the spatiotemporal scanning module.

### 3.6.2 SPATIOTEMPORAL MAMBA MODULE

To achieve a better fusion of frequency components and spatial components, the layout of 12 consecutive spatial heatmaps and the scanning strategy are the same as those in the frequency scanning branch. Then the spatiotemporal heatmap $\mathbf{F}_{st}$, which is a combination of multi spatial heatmaps $\mathbf{F}_s$, is obtained. Following the design in (Zhu et al., 2024), the Spatiotemporal Mamba Module $STScan$ is adopted to capture spatiotemporal features, formulated as

$$\mathbf{F}_s = \text{STScan}(\text{Conv}(\mathbf{F}_s)) \odot \text{SiLU}(\mathbf{F}_s), \tag{10}$$

where Conv represents a $1 \times 1$ convolution and STScan refers to the spatiotemporal Mamba mentioned earlier, the spatial Mamba module follows a similar operational sequence to the frequency Mamba module. Since the Fourier domain possesses global properties, where each pixel in the Fourier space interacts with all spatial pixels, it is natural to explore the HCP task using Fourier transforms. Therefore, the FFT is employed to transform the spatial heatmaps from the spatial domain to the Fourier domain $\mathbf{F}_s$, which encapsulates global characteristics.

Finally, the outputs from the frequency and spatiotemporal modeling branches are concatenated, followed by a $1 \times 1$ convolution to harmonize the features. An element-wise addition of a residual connection branch is then performed, yielding the final output.

### 3.7 LOSS FUNCTION

To enable joint learning across multiple tasks, we define the total loss function $L_{total}$, which consists of three components: pose estimation loss $L_{pose}$, action recognition loss $L_{action}$, and ReID loss $L_{reid}$. For $L_{pose}$, we use the Mean Per Joint Position Error (MPJPE) (Ionescu et al., 2013), which calculates the average Euclidean distance between the predicted joints $\hat{k}$ and the ground truth joints $k$. Both $L_{action}$ and $L_{reid}$ are optimized using cross-entropy loss (Zhang & Sabuncu, 2018). During evaluation, action recognition performance is assessed using accuracy and F1 score, while ReID performance is measured using mean average precision (mAP) and the cumulative matching curve (CMC) at rank-1. Therefore, the joint training loss function can be formulated as:

$$L_{total} = L_{pose} + L_{action} + L_{reid}. \tag{11}$$

Besides, we assess the model size by evaluating the total number of trainable parameters (Params).

## 4 THP DATASET

While numerous RF-based human perception datasets exist, they predominantly focus on free-space scenarios. The unique challenges posed by through-wall environments, such as signal attenuation and multipath propagation, necessitate a specialized dataset. To address this critical gap, we introduce the Through-Wall Human-Centric Perception (THP) dataset, a comprehensive dataset for the through-wall human-centric perception tasks.

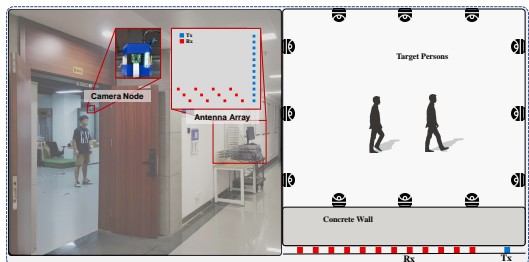

Figure 4: Experiment scenario and multi-modal dataset collection system. Rx are the receiving antennas, and Tx are the transmitting antennas.

**Data Collection System.** We designed a multi-modal data collection system, as illustrated in Figure 4, comprising two primary components. The through-wall radar system, designed for transmitting and receiving RF signals to perceive human motion, consists of a multi-input multi-output (MIMO) antenna array and a vector network analyzer (VNA). The MIMO antenna array layout is optimized to mitigate sidelobe effects, effectively forming an array of 144 virtual antennas. The VNA generates stepped-frequency continuous wave (SFCW) signals within the 0.8-2.8 GHz range, enabling a range resolution of approximately 75 mm.

To provide accurate ground truth labels, we implemented a multi-camera system with 12 nodes, each composed of a Raspberry Pi, a camera module, and an Ethernet-powered module. We employ the

calibration technique from (Zhang, 2000) to perform pairwise calibration between adjacent cameras, aligning all cameras to a unified world coordinate system.

**THP Dataset Description.** We introduce the THP dataset contains approximately 351,000 pairs of RF frames with corresponding 3D human keypoints, person identity labels, action category labels, and optical images. THP comprising two distinct subsets: THP-W (Wall-Occlusion) and THP-F (Free-space). (1) THP-W consists of 255000 pairs of RF frames collected in wall-occlusion environments, featuring 10 individuals performing 19 action categories, including both dynamic actions and static poses. The primary obstacle is a 23 cm concrete wall, presenting a significant challenge for through-wall human perception methods. (2) THP-F consists of 96000 pairs of RF frames collected in free-space environments, maintaining similar data structure and action categories as THP-W. This subset allows for evaluating model performance without wall occlusion.

Both subsets share common characteristics, including a scanning rate of 12 frames per second (FPS), single radar echo dimensions of $201 \times 144$, and optical image resolution of $1640 \times 1248$. The subjects in the dataset exhibit diverse physical characteristics, enhancing the robustness of the data. This comprehensive dataset enables rigorous evaluation of RF-based human sensing methods across various environmental conditions, from occluded to free-space scenarios, providing a valuable resource for advancing research in this field.

# 5 EXPERIMENTAL RESULTS

**Implementation Details.** All baselines and our RFMamba are trained using an Nvidia RTX4090 GPU and implemented with PyTorch. We used the Adam optimizer with an initial learning rate of 2e-3, which decays by a factor of 0.5 (gamma) every 10 epochs using the StepLR scheduler. The batch size was set to 50, and training epochs were set to 50 for all models except RadarFormer (1000 epochs due to slower convergence). For RFMamba, we stacked 12 consecutive frames as input. The dataset was split into a 4:1 training-testing ratio, using a fixed random seed of 42.

**Baselines.** To demonstrate the superiority of the proposed RFMamba framework, we compare it with various RF-based human perception baselines, including (1) RadarFormer (Zheng et al., 2023b), (2) RFPose3D (Zhao et al., 2018b), (3) mmPose (Sengupta et al., 2020), and (4) ResNet3D-50 (Hara et al., 2018). RadarFormer is the first end-to-end transformer network in through-wall scenarios. RFPose3D is the state-of-the-art(SOTA) method for human pose estimation utilizing 2D horizontal and vertical RF heatmaps. As a majority of through-wall perception methods (Zheng et al., 2023a) can be considered as variants of ResNet3D, we choose ResNet3D-50 as the backbone and MLP as the task-specific heads to perform human perception using 3D radar heatmaps.

| Method | Nose | Neck | Shoulder | Elbow | Wrist | Hip | Knee | Ankle | Eye | Ear | Mean (mm) $\downarrow$ | Params (M) |
|---|---|---|---|---|---|---|---|---|---|---|---|---|
| RF-Pose3D | 81.30 | 62.98 | 78.95 | 96.93 | 122.06 | 75.84 | 80.47 | 83.44 | 78.63 | 79.67 | 85.35 | 10.91 |
| ResNet3D-50 | 105.26 | 86.10 | 98.38 | 114.51 | 161.95 | 88.62 | 87.35 | 97.82 | 106.13 | 97.60 | 105.34 | 352.30 |
| mm-Pose | 160.92 | 147.69 | 159.82 | 189.97 | 247.54 | 145.57 | 138.19 | 140.37 | 161.65 | 156.41 | 165.98 | 22.07 |
| RadarFormer | 281.26 | 239.97 | 248.36 | 277.24 | 344.03 | 231.88 | 221.23 | 208.42 | 280.02 | 258.29 | 258.90 | 12.88 |
| RFMamba | **51.85** | **41.31** | **44.03** | **52.73** | **68.89** | **41.53** | **46.32** | **55.18** | **53.30** | **47.24** | **50.64** | 1.94 |

Table 1: Quantitative Evaluation Results for Pose Estimation Task. The notation '$\downarrow$': lower is better.

## 5.1 QUANTITATIVE RESULTS ON DIVERSE DOWNSTREAM TASKS

**Performance of Pose Estimation.** As shown in Table 1, RFMamba achieves a state-of-the-art MPJPE of 50.64 mm, significantly outperforming all baselines. This superior performance stems from three key innovations: (1) Comprehensive temporal modeling via 3D spatial-temporal volumes, surpassing RadarFormer's limited two-frame approach; (2) Advanced multi-scale feature extraction through synergistic frequency selective and spatiotemporal assistant modeling branch, addressing the temporal limitations of ResNet3D-50; and (3) Efficient information utilization via omni-dimensional frequency scanning strategy, preserving critical motion cues lost in RFPose3D's 2D heatmaps and mm-Pose's sparse point clouds. These architectural advancements enable RFMamba to effectively tackle the unique challenges of through-wall human pose estimation across diverse scenarios.

**Performance of Activity Recognition.** Table 2 presents the quantitative comparison for human activity recognition. Existing methods show limitations in temporal modeling: ResNet3D-50 relies on single-frame inputs, while RadarFormer uses only two successive frames, both resulting in suboptimal performance due to insufficient temporal information. In contrast, RFMamba significantly outperforms these baselines by employing an omni-dimensional scanning mecha-

| Method | Action Recognition | | Person Re-ID | |
|---|---|---|---|---|
| | Accuracy | F1 Score | mAP | CMC-1 |
| ResNet3D-50 | 0.9871 | 0.9820 | 0.8575 | 0.8979 |
| RadarFormer | 0.8918 | 0.7020 | 0.4817 | 0.8986 |
| RFMamba | **0.9997** | **0.9994** | **0.9991** | **0.9967** |

Table 2: Evaluation of Action Recognition and Person Re-ID.

nism that captures informative cues. Specifically, our model processes sequences of 12 frames, striking an optimal balance between inference efficiency and performance. This strategy allows RFMamba to accurately identify complex action patterns, even in challenging through-wall scenarios.

**Performance of Person ReID.** Table 2 reveals a strong correlation between ReID performance and accuracy in pose estimation and action recognition tasks. This correlation stems from the fact that individuals possess unique skeletal structures and exhibit distinctive behavioral patterns, which serve as crucial cues for ReID. Notably, RFMamba outperforms both CNN-based ResNet3D-50 and Transformer-based RadarFormer in ReID tasks. This superior performance can be attributed to RFMamba's advanced temporal modeling and feature extraction capabilities, which effectively capture these individual-specific characteristics.

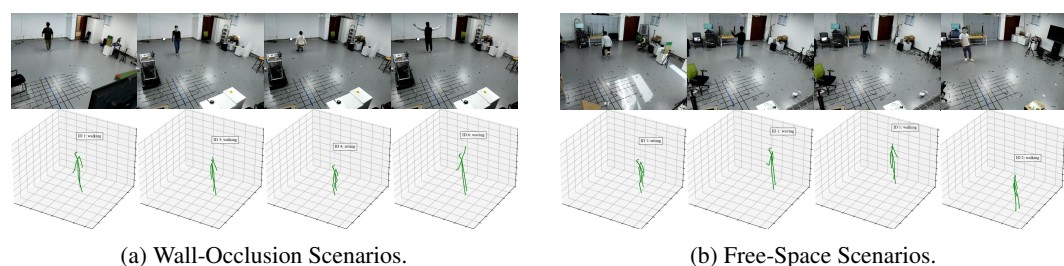

(a) Wall-Occlusion Scenarios.      (b) Free-Space Scenarios.

Figure 5: Qualitative results of 3D human pose estimation. Top row: RGB images showing various human actions. Bottom row: Corresponding 3D pose reconstructions by RFMamba.

## 5.2 QUALITATIVE ANALYSIS

**Qualitative Analysis of 3D Joint Reconstruction.** Figure 5a presents a qualitative evaluation of RFMamba in various wall-occlusive scenarios. RFMamba demonstrates superior accuracy in reconstructing detailed joint positions, particularly in challenging areas such as wrists and ankles.

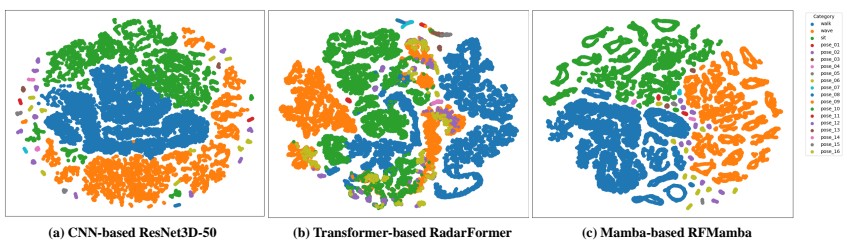

(a) CNN-based ResNet3D-50   (b) Transformer-based RadarFormer   (c) Mamba-based RFMamba

Figure 6: t-SNE visualization of learned features from different models. Each point represents a feature vector, color-coded by action category. Note the superior cluster separation and definition in RFMamba, particularly for minority classes and similar action types.

**Feature Space Visualization.** We employ t-distributed Stochastic Neighbor Embedding (t-SNE) to project decoder output features onto a 2D plane, as shown in Figure 6. RFMamba's encoder demonstrates superior feature separation, forming well-defined clusters even for minority classes and tail-end categories of our long-tailed distribution dataset. This capability is particularly evident in static pose category 16, highlighting RFMamba's ability to learn discriminative features across diverse action types. In contrast, ResNet3D-50 struggles to differentiate between dynamic scenarios like waving and walking, while RadarFormer fails to distinguish similar minority poses (e.g., static poses

5 and 14) from dynamic waving actions. These visualizations underscore RFMamba's effectiveness in capturing subtle, action-specific features, directly contributing to its superior performance in action recognition tasks.

## 5.3 GENERALIZATION CAPABILITY

To demonstrate the versatility of our approach, we evaluated its performance in free-space (unoccluded) settings using the THP-F dataset. The model achieves a MPJPE of 60.69 mm, a mAP of 1.0 and an F1 score of 1.0. The results demonstrate effective generalization to conditions different from its primary training environment. Figure 5b provides qualitative evidence of accurate pose reconstructions and action classifications in free-space environments. These results highlight the potential for applications where line-of-sight is unimpeded.

Additionally, experiments indicate that the proposed method maintains high accuracy in multi-person scenarios. Detailed results for multi-person environments are presented in Appendix A.1.

The consistent performance across varying conditions - from through-wall to free-space, and from single-person to multi-person scenarios - underscores the versatility of our approach and its potential for diverse real-world applications.

| Method | Pose | Action | Re-ID |
|---|---|---|---|
| | MPJPE (mm)$\downarrow$ | Accuracy$\uparrow$ | mAP$\uparrow$ |
| RFMamba | **50.64** | **0.9994** | **0.9991** |
| **w/o** FMB | + 9.07%(55.24) | -0.16% (0.9978) | -0.63% (0.9928) |
| **w/o** SMB | + 2.53%(51.92) | -0.13% (0.9981) | -0.61% (0.9930) |
| **w/o** FA-FFA | + 1.14%(51.22) | -0.51% (0.9943) | +0.06% (0.9997) |

Table 3: Ablation studies and analysis. The **w/o** indicates "without". FMB is the frequency modeling branch. SMB is the spatiotemporal modeling branch. FA-FFA is the FA-FFN module.

## 5.4 ABLATION STUDY.

We conduct ablation studies to evaluate the effectiveness of three key components in RFMamba. Table 3 summarizes the results across all three downstream tasks.

**Frequency Modeling Branch.** This core component of the RF-SSM block correlates RF frequency information from amplitude and phase dimensions. Removing it led to significant performance drops across all tasks, with pose estimation error increasing by 9.07%. This underscores the critical importance of frequency information modeling in RF-based tasks.

**Spatiotemporal Modeling Branch.** This branch enhances the learning process by modeling spatiotemporal features from the frequency analysis perspective. Its removal resulted in a 2.53% increase in pose estimation error, a 0.13% decrease in action recognition accuracy, and a 0.61% decrease in person ReID accuracy.

**FA-FFN Module.** The FA-FFN module adaptively identifies the most informative frequency cues, addressing the varying relevance of low and high-frequency information in human-centric perception tasks. Replacing it with a vanilla FFN led to performance degradation in pose estimation and action recognition, while maintaining similar performance in person ReID. This validates the module's effectiveness in enhancing task-specific feature extraction.

## 6 CONCLUSION

This paper introduces RFMamba, a novel approach for through-wall human sensing using RF signals. Our method incorporates an advanced architecture with omni-dimensional frequency scanning, significantly improving performance in pose estimation, action recognition, and person re-identification tasks. Comprehensive evaluations demonstrate RFMamba's superior performance and generalization capabilities across various scenarios, including free-space and multi-person environments. These advancements open new possibilities for privacy-preserving applications in healthcare, smart homes, and security systems.

ACKNOWLEDGMENTS

This work was supported by National Natural Science Foundation of China under Grants 62172381 and 62201542.

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

## A APPENDIX

In this appendix, we provide additional details and analyses to supplement our main findings on RFMamba for through-wall human-centric perception. The contents are organized as follows:

- Section A.1: Evaluation of RFMamba's performance in multi-person scenarios, including quantitative results and visual demonstrations.
- Section A.2: Detailed analysis of pose estimation performance across different action scenarios, comparing RFMamba with baseline methods.
- Section A.3: Qualitative evaluation results, presenting visual comparisons of 3D pose estimation across different methods.
- Section A.4: Analysis of action recognition performance using confusion matrices, highlighting RFMamba's capabilities in handling imbalanced data distributions.

|  | Nose | Neck | Shoulder | Elbow | Wrist | Hip | Knee | Ankle | Eye | Ear | Average |
|---|---|---|---|---|---|---|---|---|---|---|---|
| Person 1 | 142.30 | 137.61 | 149.24 | 173.12 | 211.15 | 144.49 | 143.56 | 167.17 | 145.46 | 143.11 | 157.47 |
| Person 2 | 146.84 | 117.41 | 114.59 | 122.12 | 144.50 | 125.48 | 125.47 | 132.27 | 152.49 | 118.32 | 129.71 |
| Average | 144.57 | 127.51 | 131.92 | 147.62 | 177.83 | 134.99 | 134.52 | 149.72 | 148.98 | 130.72 | 143.59 |

Table 4: Quantitative Evaluation Results of Human Body Joint Reconstruction Error (Unit: MM).

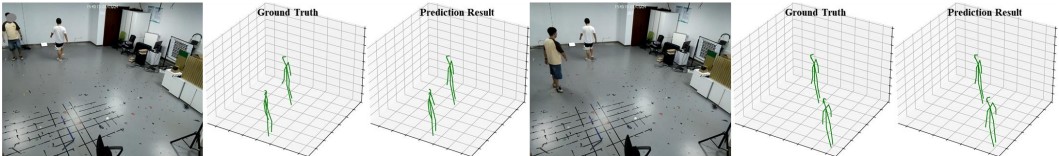

Figure 7: Qualitative Evaluation Results Under Multi-Target Scenarios.

### A.1 MULTI-PERSON SCENARIO EVALUATION

To comprehensively evaluate RFMamba's performance in multi-person scenarios, we conducted experiments with both two-person and three-person interactions. We expanded RFMamba's output dimensionality to accommodate different numbers of targets in multi-person prediction.

#### A.1.1 TWO-PERSON INTERACTION

We first collected a dataset of approximately 35,000 pairs of RF frames featuring two-person interactions. This dataset includes corresponding optical images, action labels, and human keypoints extracted using AlphaPose. Our evaluation focused on a challenging scenario where Person 1 performs walking while Person 2 executes a marching-in-place action. This setup captures complex spatial displacements and joint dynamics.

Table 4 presents the pose reconstruction errors for this two-person scenario. While there is a slight increase in reconstruction error compared to single-person scenarios, RFMamba maintains competitive performance. Person 1 exhibits higher reconstruction errors due to increased motion complexity.

Figure 7 illustrates RFMamba's effective simultaneous reconstruction of two targets. These findings demonstrate RFMamba's capability to handle basic multi-person interactions effectively.

#### A.1.2 THREE-PERSON INTERACTION

To further validate the generalization capability, we extended our evaluation to more complex scenarios with three individuals. We collected a comprehensive dataset comprising 48000 pairs of RF frames across different temporal periods, environments, and occlusion conditions (with/without

wooden board obstruction). The dataset includes corresponding optical images, action categories, and human keypoints extracted using AlphaPose. The data was split into a training set and a test set at a 4:1 ratio.

The individuals engaged in three complex dynamic activities simultaneously: walking freely, waving hands, and transitioning between standing and sitting positions. This setup provides a more challenging test of the model's capabilities in handling multiple targets with diverse motions.

|  | Nose | Neck | Shoulder | Elbow | Wrist | Hip | Knee | Ankle | Eye | Ear | Average |
|---|---|---|---|---|---|---|---|---|---|---|---|
| Person 1 | 78.56 | 79.15 | 81.8 | 111.56 | 170.25 | 79.96 | 81.94 | 82.31 | 77.94 | 79.58 | 93.80 |
| Person 2 | 50.49 | 39.33 | 40.97 | 41.13 | 46.22 | 36.63 | 36.14 | 35.86 | 49.87 | 51.37 | 42.57 |
| Person 3 | 86.70 | 81.95 | 86.24 | 100.87 | 125.49 | 82.98 | 85.81 | 93.32 | 86.20 | 87.35 | 92.51 |
| Average | **71.92** | **66.81** | **69.67** | **84.52** | **113.99** | **66.52** | **67.96** | **70.5** | **71.34** | **72.77** | **76.29** |

Table 5: Pose reconstruction errors (mm) in three-person scenario without wooden board obstruction.

|  | Nose | Neck | Shoulder | Elbow | Wrist | Hip | Knee | Ankle | Eye | Ear | Average |
|---|---|---|---|---|---|---|---|---|---|---|---|
| Person 1 | 108.60 | 94.01 | 99.90 | 116.17 | 140.15 | 93.66 | 93.48 | 102.19 | 108.76 | 102.82 | 106.49 |
| Person 2 | 67.06 | 35.24 | 38.23 | 65.89 | 124.37 | 31.97 | 33.23 | 37.06 | 68.14 | 51.54 | 55.73 |
| Person 3 | 96.44 | 94.37 | 94.92 | 96.09 | 99.82 | 90.31 | 81.25 | 78.33 | 98.59 | 99.05 | 92.64 |
| Average | **90.7** | **74.54** | **77.68** | **92.72** | **121.45** | **71.98** | **69.32** | **72.53** | **91.83** | **84.47** | **84.95** |

Table 6: Pose reconstruction errors (mm) in three-person scenario with wooden board obstruction.

Tables 5 and 6 present the pose reconstruction errors for three-person scenarios without and with wooden board obstruction, respectively. Despite the increased complexity, RFMamba maintains robust performance across different individuals and occlusion conditions. The variations in error rates can be attributed to different motion complexities and occlusion effects.

Figure 8 demonstrates RFMamba's effective simultaneous reconstruction of three targets in both unobstructed and obstructed scenarios. These comprehensive results further support RFMamba's potential for real-world applications involving multiple subjects.

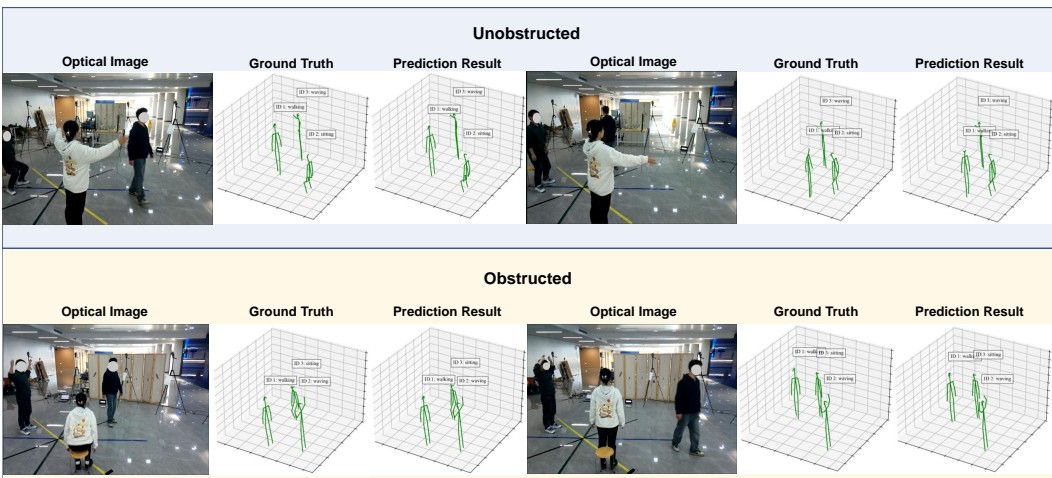

Figure 8: Qualitative results of three-person pose estimation in both unobstructed and obstructed scenarios. For each case: optical image (left), ground truth pose (middle), and RFMamba prediction (right).

## A.2 POSE ESTIMATION PERFORMANCE ACROSS DIFFERENT ACTION SCENARIOS

We further provide a detailed analysis of RFMamba's pose estimation performance across various action scenarios, further illustrating its capabilities in diverse conditions. Table 7 presents a comparative analysis of different methods for actions including walking, waving, sitting, and static poses.

| Method | Walk | Wave | Sit | Static | Mean |
|---|---|---|---|---|---|
| RF-Pose3D | 114.28 | 70.46 | 65.43 | 46.46 | 85.35 |
| mm-Pose | 225.69 | 110.21 | 141.00 | 132.88 | 165.98 |
| ResNet3D-50 | 150.10 | 74.91 | 76.54 | 74.28 | 105.34 |
| RadarFormer | 443.87 | 117.09 | 164.17 | 86.26 | 258.90 |
| RFMamba | **72.08** | **34.81** | **38.97** | **31.17** | **50.64** |

Table 7: Detailed MPJPE (mm) Under Different Action Scenarios.

The results show that RFMamba consistently achieves lower reconstruction errors compared to existing methods across all action categories. In particular, for challenging scenarios such as walking, which involve complex non-rigid motions and significant variations in posture and position, RFMamba maintains relatively low reconstruction errors. This performance indicates the model's ability to accurately capture and reconstruct human poses even in dynamic and complex environments.

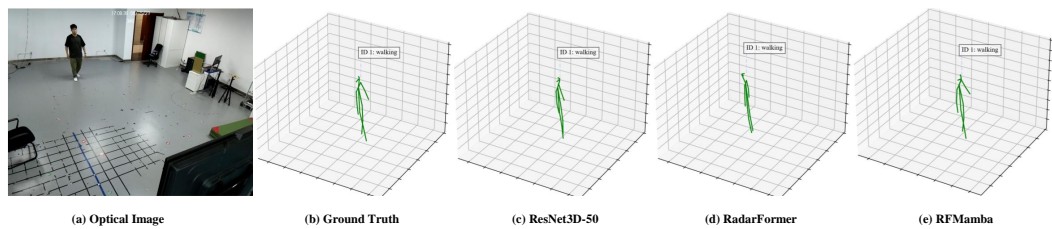

Figure 9: Qualitative comparison of 3D pose estimation results: (a) Optical image for reference, (b) Ground truth pose, and estimated poses by (c) ResNet3D-50, (d) RadarFormer, and (e) RFMamba.

### A.3 QUALITATIVE EVALUATION RESULTS OF DIFFERENT METHODS

Figure 9 provides a visual comparison of 3D pose estimation results from RFMamba and baseline methods to complement the quantitative evaluations. These examples illustrate RFMamba's improved accuracy in pose reconstruction and temporal consistency across frames. While ResNet3D-50 shows overall effectiveness, it exhibits notable deviations in extremities. RadarFormer, limited by its simpler network structure and narrow temporal receptive field, struggles with complex poses, only approximating body positions rather than precise joint locations. These visual results underscore RFMamba's ability to capture fine-grained spatial information and temporal dynamics.

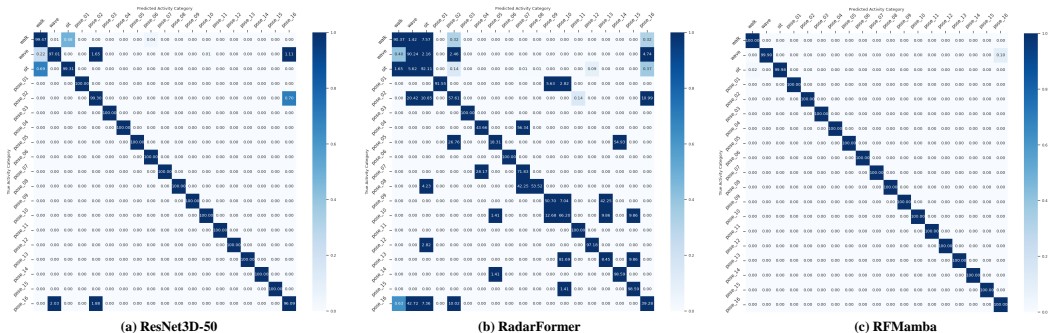

Figure 10: Comparison of confusion matrices for activity recognition task: (a) ResNet3D-50, (b) RadarFormer, and (c) RFMamba. Darker colors indicate higher prediction accuracy.

A.4   EVALUATING REID PERFORMANCE WITH CONFUSION MATRIX

Figure 10 presents the average confusion matrices for ResNet3D-50, RadarFormer, and RFMamba, providing insights into their action recognition performance. RFMamba demonstrates superior accuracy across various actions, including those with fewer samples in the long-tailed data distribution.

ResNet3D-50, which infers action classes from single frames, shows a bias towards classifying minority samples as the majority class (walking). RadarFormer performs well in distinguishing dynamic actions but struggles with minority classes due to insufficient temporal information.

In contrast, RFMamba accurately recognizes both majority and minority classes, showcasing its ability to learn effective classification boundaries and its strong generalization capability, even with imbalanced data distributions.

