# OpenReview forum: "RFMamba: Frequency-Aware State Space Model for RF-Based Human-Centric Perception"
_ICLR.cc/2025/Conference — ICLR 2025 Poster_

### Official Review · Reviewer_aiy3 · 2024-10-29

**Soundness:** 2
**Presentation:** 3
**Contribution:** 3
**Rating:** 8
**Confidence:** 4

**Summary:**

The paper presents RFMamba, a novel frequency-aware state space model for RF-based human-centric perception (HCP), designed to process radio frequency (RF) signals for tasks like human pose estimation, activity recognition, and person re-identification (ReID). Leveraging the capabilities of state space models (SSMs), RFMamba uses a dual-branch architecture that combines frequency and spatiotemporal modeling to effectively capture the characteristics of RF signals. The model introduces an omni-dimensional scanning mechanism, selectively focusing on informative frequency cues. Experiments on the newly introduced THP dataset (covering both free-space and wall-occlusion scenarios) demonstrate RFMamba’s superior performance over existing methods in accuracy.

**Strengths:**

1. Motivation and practical use cases: The authors address real-world challenges in RF-based human perception, aiming to support applications like security, health monitoring, and emergency response by enabling through-wall sensing and multi-person tracking where visual methods may be ineffective.

2. Technical novelty with Mamba modules: RFMamba leverages Mamba-based state space modeling specifically adapted for RF signals, incorporating frequency and spatiotemporal Mamba modules to capture long-term dependencies within complex RF data.

3. Generally clear presentation: The paper is well-structured and understandable, though some details are missing, particularly regarding some design choices, data collection and experimental setup (see weaknesses below).

**Weaknesses:**

1. Lack of sequence length analysis: While the authors emphasize the model's capability to handle variable and long input sequences, they do not provide an analysis of how sequence length affects performance, efficiency, or stability. It would be useful to see experiments with varying sequence lengths, insights into computational complexity and memory impact, and clarification on the sequence length used in their experiments.
2. Potential for improved transformer-based model: The lower performance of RadarFormer likely reflects specific design choices rather than a fundamental transformer limitation for RF perception (e.g., using only amplitudes, using two successive frames instead of 12). The authors did not explore enhancements to RadarFormer, such as incorporating phase information, extending the temporal context, or adding RF-specific adaptations, which may have yielded results comparable to or better than RFMamba, questioning the necessity of Mamba-based SSMs.
3. Unclear design choices: Some design decisions are not fully explained. For instance, in Figure 3(a), the scanning path appears to follow a bidirectional approach converging at the center. It is unclear why the authors did not consider capturing both sequences (from T−6 to T+6 and T+6 to T−6) and concatenating them, which could potentially enrich the temporal representation and improve performance.
4. Lack of details on activity types, distribution, and environment diversity: The paper lacks specifics on the 19 activities in the THP dataset, their frequency distribution, and whether participants performed these activities statically or while moving freely. This is important for understanding the model’s generalization across activity types and movement patterns in RF-based perception. Additionally, the dataset was collected primarily in a single environment (with and without a wall), limiting environmental diversity. It remains unclear if the model would perform similarly in different settings, whether it would need extensive data collection, or if fine-tuning would be required in new environments.
5. Limited dataset accessibility: Although the THP dataset provides valuable scenarios for evaluating RF-based human perception, it is not confirmed for public release, restricting reproducibility and broader research applications in this area.
6. Unclear real-time performance: The paper lacks analysis on the compute and memory requirements needed for real-time performance, leaving practical deployment feasibility uncertain.
7. Lack of generalization comparison with baseline models: While the authors claim that RFMamba generalizes well across various scenarios, there is no direct comparison with existing methods on generalization performance. Without showing how baseline models perform in similar conditions, it is difficult to assess whether RFMamba’s improvements are specific to the training environment or if they extend to varied, unseen scenarios.

**Questions:**

1. What are the specific activities in the THP dataset, and what is their distribution?
2. Were participants stationary or moving freely during activities?
3. Would the model generalize to other environments without additional data collection or fine-tuning?
4. The authors used SFCW radars for evaluation. Would the proposed method generalize to FMCW radar systems as well, and if so, what adjustments might be necessary?
5. Will the THP dataset be made publicly available to support reproducibility and further research in RF-based human perception?
6. What are the compute and memory requirements for achieving real-time performance with RFMamba?
7. The dataset was split into a 4:1 training-testing ratio. How much data was allocated for validation, and what criteria were used for this split?
8. For pose estimation, the authors used the metric Mean Per Joint Position Error (MPJPE). Is this measured in 3D or 2D space?
9. Was IRB approval obtained for the human subject data collection, and were participants informed and provided consent for the use of their radar and camera data in this study?

**Details Of Ethics Concerns:**

The authors mentioned the approval of IRB and receiving consent of the subjects in the rebuttal. No more concerns regarding this.

---

> ### Author Response · Authors · 2024-11-23
>
> Thank you for the valuable suggestions and insights, which have significantly improved our manuscript.
>
> ------------
>
> **Q1 Lack of sequence length analysis: While the authors emphasize the model's capability to handle variable and long input sequences, they do not provide an analysis of how sequence length affects performance, efficiency, or stability. It would be useful to see experiments with varying sequence lengths, insights into computational complexity and memory impact, and clarification on the sequence length used in their experiments.**
>
> Thank you for your insightful suggestion. To address this, we conducted comprehensive experiments to analyze the impact of sequence length on our model's performance, efficiency, and stability. The results are summarized in the table below:
>
> | Sequence Length | MPJPE | Accuracy | mAP    | Complexity (GFLOPS) | Memory (Parameters, MB) |
> | --------------- | ----- | -------- | ------ | ------------------- | ----------------------- |
> | 1-frame         | 80.69 | 0.9965   | 0.9721 | 0.0382              | 1.66                    |
> | 2-frame         | 70.10 | 0.9974   | 0.9823 | 0.0752              | 1.68                    |
> | 6-frame         | 65.84 | 0.9988   | 0.9968 | 0.2234              | 1.76                    |
> | 12-frame        | 50.64 | 0.9994   | 0.9991 | 0.4458              | 1.94                    |
> | 18-frame        | 50.14 | 0.9995   | 0.9993 | 0.6682              | 2.09                    |
> | 24-frame        | 51.04 | 0.9994   | 0.9992 | 0.8905              | 2.21                    |
>
> The results indicate that a sequence length of 12 frames provides optimal performance in terms of MPJPE, achieving a good balance between accuracy, computational complexity, and memory usage. Shorter sequences (1-6 frames) lead to reduced performance, while longer sequences (18 frames) show diminishing performance improvements at the cost of significantly increased computational requirements.
>
> ------------------
> **Q2 Potential- for improved transformer-based model: The lower performance of RadarFormer likely reflects specific design choices rather than a fundamental transformer limitation for RF perception (e.g., using only amplitudes, using two successive frames instead of 12). The authors did not explore enhancements to RadarFormer, such as incorporating phase information, extending the temporal context, or adding RF-specific adaptations, which may have yielded results comparable to or better than RFMamba, questioning the necessity of Mamba-based SSMs.**
>
> We sincerely appreciate your insightful comment. Based on your suggestions, we implemented several enhancements to RadarFormer to further explore its potential:
>
> 1. Extended the temporal context to 12-frame sequences (matching RFMamba).
> 2. Incorporated both amplitude and phase information.
> 3. Modified the cross-attention mechanism to consider an 11-frame history window.
> 4. Integrated RF-specific adaptations inspired by RFMamba.
>
> Due to the slower convergence of the enhanced RadarFormer, we report its performance after 80 epochs (compared to 50 epochs for our RFMamba) as follows:
>
>
> | Model                | MPJPE (mm) | Accuracy   | mAP        |
> | -------------------- | ---------- | ---------- | ---------- |
> | Original RadarFormer | 258.90     | 0.8918     | 0.4817     |
> | Enhanced RadarFormer | 90.85      | 0.9931     | 0.9598     |
> | **Our RFMamba**      | **50.64**  | **0.9994** | **0.9991** |
>
> While the enhanced RadarFormer shows significant improvement, RFMamba still outperforms it across all metrics. To further evaluate the necessity of Mamba-based SSMs, we conducted an additional experiment by replacing the SSM module with a multi-head attention mechanism while keeping the rest of the architecture unchanged. The results are as follows:
>
> | Model           | Batch Size | GPU Memory Usage | MPJPE (mm) | Accuracy | mAP    |
> | --------------- | ---------- | ---------------- | ---------- | -------- | ------ |
> | **Mamba-based** | 50         | 6.13 GB          | 50.64      | 0.9994   | 0.9991 |
> | Attention-based | 30         | 23.3 GB          | 55.22      | 0.9985   | 0.9972 |
>
> These results demonstrate that the Mamba-based model achieves superior performance with significantly lower memory requirements, highlighting its efficiency and effectiveness for RF perception tasks where both performance and resource constraints are critical.
>
> --------------------

---

> > ### Author Response · Authors · 2024-11-23
> >
> > **Q3 Unclear design choices: Some design decisions are not fully explained. For instance, in Figure 3(a), the scanning path appears to follow a bidirectional approach converging at the center. It is unclear why the authors did not consider capturing both sequences (from T−6 to T+6 and T+6 to T−6) and concatenating them, which could potentially enrich the temporal representation and improve performance.**
> >
> > Thank you for raising this important question. Our scanning strategy—starting from T−6 (−0.5s) and T+6 (0.5s) and converging at the center (0s)—is designed based on the temporal relevance of neighboring frames to the current frame. As shown in **Figure 11 within the Appendix**, the contribution distribution of neighboring frames to the current frame reveals a Gaussian-like pattern, with the following observations:
> >
> > - The highest contribution occurs at the current moment (0s), and it decreases gradually as the temporal distance increases.
> > - Heatmaps illustrate that frames closer to the current time (e.g., −0.5s, 0s, and 0.5s) provide significantly more information for pose estimation compared to frames further away.
> >
> > This observation aligns with the temporal locality of RF signals, where the most relevant information for estimating the current motion state is concentrated within a roughly 1-second range centered around the current frame. By designing a scanning strategy that converges at 0s from both T−6 (−0.5s) and T+6 (0.5s), the model effectively focuses on the most meaningful temporal segments, avoiding redundant or less relevant information from reverse-order sequences.  We believe this approach ensures optimal use of the temporal context for RF human perception.
> >
> > In contrast, capturing and concatenating both sequences (T−6 to T+6 and T+6 to T−6) assumes equal importance for all temporal frames, potentially diluting the model's focus on the key timeframes around 0s.
> >
> > To validate our design, we conducted an experiment comparing our strategy with an alternative approach that captures and concatenates both sequences (T−6 to T+6 and T+6 to T−6). The results are shown below:
> >
> > | Scanning Strategy | Pose Estimation (MPJPE ) | Person Re-ID (mAP) | Action Recognition (Accuracy) |
> > | ----------------- | ------------------------ | ------------------ | --------------- |
> > | Our Strategy      | 50.64                    | 0.9991             | 0.9997                        |
> > | Both Sequences    | 53.08                    | 0.9883             | 0.9992                        |
> >
> > These results indicate that our proposed scanning strategy outperforms the alternative in all three tasks, validating the design choice. By focusing on the most relevant temporal frames around 0s, our method avoids introducing redundant or less meaningful information. We hope this explanation addresses your concern, and we welcome further feedback or suggestions for additional exploration.
> >
> > ----
> > **Q4 Lack of details on activity types, distribution, and environment diversity: The paper lacks specifics on the 19 activities in the THP dataset, their frequency distribution, and whether participants performed these activities statically or while moving freely. This is important for understanding the model’s generalization across activity types and movement patterns in RF-based perception. Additionally, the dataset was collected primarily in a single environment (with and without a wall), limiting environmental diversity. It remains unclear if the model would perform similarly in different settings, whether it would need extensive data collection, or if fine-tuning would be required in new environments.**
> >
> > **(What are the specific activities in the THP dataset, and what is their distribution?)**
> >
> > **(Were participants stationary or moving freely during activities?)**
> >
> > **(Would the model generalize to other environments without additional data collection or fine-tuning?)**
> >
> > Our dataset consists of 19 activity types, including 3 dynamic actions and 16 static poses.
> >
> > The dynamic actions were designed to capture varying levels of movement complexity:
> >
> > - **Walking**: Randomly walking in the environment, representing higher action complexity with significant variations in posture and position.
> > - **Waving**: Focuses on fine-grained limb movements while the torso remains stationary.
> > - **Sitting-to-Standing Transitions**: Involves substantial changes in the torso's position and posture, reflecting coarse body movements.
> >
> > In addition, we included 16 common static poses to evaluate the model's ability to perceive and distinguish various static poses. These poses encompass a wide range of body postures, enhancing the dataset's comprehensiveness.
> >
> > As shown in **Figure 12 within the Appendix**, dynamic actions account for 93.1% of the dataset, emphasizing complex movement scenarios. Static poses, though comprising only 6.9%, are included to provide a complementary evaluation of the model’s ability to generalize across less frequent activity types.

---

> > > ### Author Response · Authors · 2024-11-23
> > >
> > > To evaluate the model's generalization to a new environment, we conducted experiments in a new environment where we collected approximately 24000 pairs of RF frames, corresponding optical images, action categories, and human keypoints extracted using AlphaPose. The data was split into a training set (18000 pairs) and a test set (6000 pairs) at a 4:1 ratio, with 3000 pairs from the training set used for fine-tuning.
> > >
> > > We evaluated two approaches: (1) training the network from scratch with the full 18000 pairs, and (2) fine-tuning the pre-trained model with only the 3000 pairs. The model's performance on the test set is summarized as follows:
> > >
> > > | **Approach** | Extensive Training | Fine-tuning |
> > > | ------------ | ------------------ | ----------- |
> > > | MPJPE (mm)   | 76.64              | 97.73       |
> > >
> > > These results indicate that extensive training provides better accuracy, but fine-tuning achieves reasonable performance with significantly less data. This demonstrates the model’s ability to adapt to new environments effectively, depending on the data availability and desired accuracy.
> > >
> > > ----------------------
> > >
> > > **Q5 Limited dataset accessibility: Although the THP dataset provides valuable scenarios for evaluating RF-based human perception, it is not confirmed for public release, restricting reproducibility and broader research applications in this area.**
> > >
> > > (**Will the THP dataset be made publicly available to support reproducibility and further research in RF-based human perception?)**
> > >
> > > Thank you for highlighting this important concern. We fully recognize that dataset accessibility is vital for ensuring reproducibility and advancing research in RF-based human perception. We are pleased to share that we are actively preparing the THP dataset for public release, with the release planned within the next three months. The public release will include not only the dataset but also comprehensive documentation and usage guidelines to facilitate its adoption by the research community.
> > >
> > > ----------------------
> > >
> > > **Q6 Unclear real-time performance: The paper lacks analysis on the compute and memory requirements needed for real-time performance, leaving practical deployment feasibility uncertain.**
> > >
> > > **(What are the compute and memory requirements for achieving real-time performance with RFMamba?)**
> > >
> > > Thank you for highlighting the importance of real-time performance analysis.we evaluated RFMamba’s computational and memory efficiency on an NVIDIA RTX 4090 GPU. The results are summarized below:
> > >
> > > | Metric          | MPJPE (mm) | Accuracy | mAP    | Parameters (M) | Throughput (FPS) | FLOPS (GFLOPS) |
> > > | --------------- | ---------- | -------- | ------ | -------------- | ---------------- | -------------- |
> > > | **Our RFMamba** | 50.64      | 0.9994   | 0.9991 | 1.94           | 60.60            | 0.4458         |
> > >
> > > RFMamba achieves **real-time performance at 60.60 FPS** while maintaining high accuracy (MPJPE: 50.64 mm). With its lightweight architecture of 1.94M parameters and 0.4458 GFLOPS, RFMamba is highly efficient, making it suitable for real-world deployment scenarios.
> > >
> > > ----------------------------
> > > **Q7 Lack of generalization comparison with baseline models: While the authors claim that RFMamba generalizes well across various scenarios, there is no direct comparison with existing methods on generalization performance. Without showing how baseline models perform in similar conditions, it is difficult to assess whether RFMamba’s improvements are specific to the training environment or if they extend to varied, unseen scenarios.**
> > >
> > > Thank you for highlighting the importance of comparing RFMamba’s generalization performance with baseline models. To address this, we conducted a comprehensive evaluation of several baseline methods, including RF-Pose3D, ResNet3D-50, and RadarFormer.

---

> ### Author Response · Authors · 2024-11-23
>
> - **Performance Under Non-occluded Scenarios**
>
> We further conduct the comparison on the non-occluded scenarios in the THP-F (free-space) dataset. The results for pose estimation performance in non-occluded scenarios are summarized below:
>
> | Method      | Nose      | Neck      | Shoulder  | Elbow     | Wrist     | Hip       | Knee      | Ankle     | Eye       | Ear       | Mean      |
> | ----------- | --------- | --------- | --------- | --------- | --------- | --------- | --------- | --------- | --------- | --------- | --------- |
> | RF-Pose3D   | 75.96     | 65.89     | 71.70     | 91.98     | 132.56    | 66.05     | 68.81     | 83.34     | 78.14     | 72.33     | 81.76     |
> | mm-Pose     | 115.57    | 100.09    | 126.48    | 166.45    | 206.34    | 109.53    | 111.29    | 114.25    | 115.85    | 108.76    | 129.64    |
> | ResNet3D-50 | 136.96    | 109.70    | 128.53    | 127.47    | 159.77    | 113.47    | 129.56    | 131.46    | 132.44    | 131.76    | 130.86    |
> | RadarFormer | 213.33    | 202.38    | 205.40    | 221.46    | 262.95    | 202.95    | 196.39    | 197.58    | 214.63    | 207.28    | 212.94    |
> | **RFMamba** | **63.94** | **54.61** | **56.37** | **63.53** | **83.44** | **53.81** | **50.34** | **56.02** | **63.84** | **59.60** | **60.69** |
>
> RFMamba consistently outperforms baseline methods in non-occluded scenarios, achieving significantly lower reconstruction errors across all keypoints. This demonstrates its superior capability in capturing precise human poses.
>
> - **Performance Under Different Action Scenarios**
>
> To further evaluate generalization across varied action categories, we also analyzed the performance of these methods in walking, waving, sitting, and static postures. The results are presented below:
>
> | Method      | Walk      | Wave      | Sit       | Static    | Mean      |
> | ----------- | --------- | --------- | --------- | --------- | --------- |
> | RF-Pose3D   | 114.28    | 70.46     | 65.43     | 46.46     | 85.35     |
> | mm-Pose     | 225.69    | 110.21    | 141.00    | 132.88    | 165.98    |
> | ResNet3D-50 | 150.10    | 74.91     | 76.54     | 74.28     | 105.34    |
> | RadarFormer | 443.87    | 117.09    | 164.17    | 86.26     | 258.90    |
> | **RFMamba** | **72.08** | **34.81** | **38.97** | **31.17** | **50.64** |
>
> RFMamba achieves the lowest reconstruction errors across all action scenarios, particularly excelling in challenging dynamic actions like walking, which involve significant non-rigid motion and posture variations. This highlights its ability to generalize effectively to diverse and unseen scenarios.
>
> These results, along with detailed analyses for various scenarios provided in **Appendix A.2**, demonstrate that RFMamba’s improvements are not limited to the training environment but extend robustly across different conditions and action types. We will ensure that these comparisons are clearly presented in the revised paper to address your concern.
>
> - **Performance Under Varied and Unseen Scenarios**
>
> To further validate the generalization capability, we extended our evaluation to a varied and unseen scenarios with three individuals. We collected a comprehensive dataset comprising 48000 pairs of RF frames across different temporal periods, environments, and occlusion conditions (with/without wooden board obstruction). The data includes corresponding optical images, action categories, and human keypoints extracted using AlphaPose. The data was split into a training set and a test set at a 4:1 ratio.The individuals engaged in three complex dynamic activities simultaneously: walking freely, waving hands, and transitioning between standing and sitting positions. This setup provides a more challenging test of the model's capabilities in handling multiple targets with diverse motions.
>
> > **Pose Reconstruction Errors (mm) in Three-Person Scenario Without Wooden Board Obstruction**
>
> |              | Nose      | Neck      | Shoulder  | Elbow     | Wrist      | Hip       | Knee      | Ankle     | Eye       | Ear       | Average   |
> | ------------ | --------- | --------- | --------- | --------- | ---------- | --------- | --------- | --------- | --------- | --------- | --------- |
> | **Person 1** | 78.56     | 79.15     | 81.80     | 111.56    | 170.25     | 79.96     | 81.94     | 82.31     | 77.94     | 79.58     | 93.80     |
> | **Person 2** | 50.49     | 39.33     | 40.97     | 41.13     | 46.22      | 36.63     | 36.14     | 35.86     | 49.87     | 51.37     | 42.57     |
> | **Person 3** | 86.70     | 81.95     | 86.24     | 100.87    | 125.49     | 82.98     | 85.81     | 93.32     | 86.20     | 87.35     | 92.51     |
> | **Average**  | **71.92** | **66.81** | **69.67** | **84.52** | **113.99** | **66.52** | **67.96** | **70.50** | **71.34** | **72.77** | **76.29** |

---

> ### Author Response · Authors · 2024-11-23
>
> > **Pose Reconstruction Errors (mm) in Three-Person Scenario With Wooden Board Obstruction**
>
> |              | Nose      | Neck      | Shoulder  | Elbow     | Wrist      | Hip       | Knee      | Ankle     | Eye       | Ear       | Average   |
> | ------------ | --------- | --------- | --------- | --------- | ---------- | --------- | --------- | --------- | --------- | --------- | --------- |
> | **Person 1** | 108.60    | 94.01     | 99.90     | 116.17    | 140.15     | 93.66     | 93.48     | 102.19    | 108.76    | 102.82    | 106.49    |
> | **Person 2** | 67.06     | 35.24     | 38.23     | 65.89     | 124.37     | 31.97     | 33.23     | 37.06     | 68.14     | 51.54     | 55.73     |
> | **Person 3** | 96.44     | 94.37     | 94.92     | 96.09     | 99.82      | 90.31     | 81.25     | 78.33     | 98.59     | 99.05     | 92.64     |
> | **Average**  | **90.70** | **74.54** | **77.68** | **92.72** | **121.45** | **71.98** | **69.32** | **72.53** | **91.83** | **84.47** | **84.95** |
>
> These results demonstrate that RFMamba's improvements extend beyond the training environment to varied, unseen scenarios, showing robust generalization across different individuals, environments, and occlusion conditions.
>
> Additionally, we provide the qualitative evaluation results under multi-target scenarios in the  **Figure 8 within the Appendix**, which demonstrate that the proposed algorithm can effectively achieve simultaneous reconstruction in multi-target scenarios.
>
> ---------------
>
> **Q8 The authors used SFCW radars for evaluation. Would the proposed method generalize to FMCW radar systems as well, and if so, what adjustments might be necessary?**
>
> **HIBER Dataset :** HIBER [1] is a public FMCW mmWave radar human perception dataset that includes a diverse range of environments, users, occlusions, and actions. We evaluated RFMamba on this dataset and compared with state-of-the-art FMCW mmWave-based methods. The evaluation results are summarized in the following table.
>
> | Model       | MPJPE (mm) | Params (M) | GFLOPs   |
> | ----------- | ---------- | ---------- | -------- |
> | RF-Pose3D   | 136.8      | 9.492      | 39.58    |
> | mmPose      | 102.6      | 33.08      | 0.38     |
> | RPM         | 59.20      | 81.67      | 2148.44  |
> | **RFMamba** | **61.44**  | **2.29**   | **1.48** |
>
> RFMamba achieves a performance of 61.44 mm MPJPE while utilizing only 2.29M parameters and 1.48 GFLOPs. This is comparable to the state-of-the-art model RPM [2], which achieves 59.20 mm MPJPE but requires 81.67M parameters and 2148.44 GFLOPs. These findings underscore RFMamba's strong generalization capabilities across various signal types and domains.
>
> **Adjustment Strategies :** To demonstrate the applicability of our model to FMCW radar-based datasets, we take the HIBER dataset as an illustrative example. The HIBER dataset provides data in the format (Frame, Adc_Sample, TxTr), where "Frame" represents consecutive radar echoes, "Adc_Sample" refers to the ADC sampling rate, and "TxTr" denotes the number of transmit-receive antenna pairs. These components align well with RFMamba’s omni-dimensional scanning mechanism: "Frame" leverages RFMamba’s capability to model the slow-time dimension, "Adc_Sample" corresponds to its processing of the fast-time dimension, and "TxTr" integrates seamlessly with its antenna-channel scanning capabilities. By modifying the task-specific prediction head, RFMamba can further adapt to various downstream tasks. This adaptability suggests that RFMamba can effectively generalize to FMCW systems with minimal modifications.
>
> ---------------
>
> **Q9 The dataset was split into a 4:1 training-testing ratio. How much data was allocated for validation, and what criteria were used for this split?**
>
> Thank you for pointing this out. In our current experimental setup, the dataset was split into a 4:1 training-testing ratio, and we did not allocate a separate validation set. The primary reason for this choice was to maximize the amount of data available for training, while ensuring sufficient data for robust testing.
>
> -------------
>
> **Q10 For pose estimation, the authors used the metric Mean Per Joint Position Error (MPJPE). Is this measured in 3D or 2D space?**
>
> Thank you for your question. The Mean Per Joint Position Error (MPJPE) used in our experiments is measured in 3D space. It calculates the Euclidean distance between the predicted and ground truth joint positions in 3D coordinates.
>
>
> ----------------

---

> > ### Author Response · Authors · 2024-11-24
> >
> > **Q11 Was IRB approval obtained for the human subject data collection, and were participants informed and provided consent for the use of their radar and camera data in this study?**
> >
> > Thank you for your question.
> >
> > - **IRB Approval**: This study was conducted under Institutional Review Board (IRB) approval to ensure compliance with ethical guidelines for human subject research.
> > - **Informed Consent**: All participants were fully informed about the purpose of the study and the use of their radar and camera data, and they provided explicit written consent prior to their participation.
> >
> > -------------------------
> >
> > **References**
> > [1] Wu, Z., Zhang, D., Xie, C., Yu, C., Chen, J., Hu, Y., and Chen, Y., "RFMask: A simple baseline for human silhouette segmentation with radio signals," *IEEE Transactions on Multimedia*, vol. 25, pp. 4730–4741, 2022.
> >
> > [2] Xie, C., Zhang, D., Wu, Z., Yu, C., Hu, Y., and Chen, Y., "RPM: RF-Based Pose Machines," *IEEE Transactions on Multimedia*, vol. 26, pp. 637–649, 2024, doi: 10.1109/TMM.2023.3268376.

---

> ### Author Response · Authors · 2024-11-25
>
> We sincerely appreciate Reviewer Aiy3's recognition of our revisions and the thoughtful decision to raise the evaluation of our work from 5 to 8. Your valuable suggestions have significantly improved the quality of our paper.

---

> > ### Comment · Reviewer_aiy3 · 2024-12-02
> >
> > Thank you for carefully considering my feedback. The revised manuscript successfully addresses all of my concerns.

---

> > > ### Author Response · Authors · 2024-12-03
> > >
> > > We are truly grateful for your valuable feedback and delighted that our revisions have successfully addressed all your concerns.  Thank you for your professionalism and constructive insights throughout the review process.

---

### Official Review · Reviewer_5Q2S · 2024-11-02

**Soundness:** 3
**Presentation:** 3
**Contribution:** 3
**Rating:** 6
**Confidence:** 3

**Summary:**

This paper proposes a State Space Model (SSM) based method for human-centric perception with radio frequency (RF) signals. Specifically, the authors devise a two-branch model to learn frequency and spatial-temporal features, respectively. The authors have also conducted comprehensive experiments to demonstrate the superiority of their method.

**Strengths:**

1. The use of Mamba models in RF-based sensing is novel.
2. The authors have collected a new dataset and evaluated their method with three downstream tasks.

**Weaknesses:**

1. The motivation of using Mamba models is not clear enough. The authors mention that Mamba models are more capable of tackling long sequences, while the authors have not discussed the performance of models in terms of different lengths of sequences.
2. A common challenge in RF-based sensing is cross-domain sensing, since RF signals may differ between different environments. I understand that this paper pioneers the use of Mamba models for RF-based sensing, rather than focusing on cross-domain sensing, but the authors should provide more discussions on this.
3. The authors have compared their method with baselines in terms of the number of parameters, while the authors can also compare their detailed efficiency, such as training time, testing time, and throughput.
4. Many related works about Wireless Signal Sensing have not been discussed. For example, recent works have involved the use of WiFi signals, ultra-wideband radars, millimeter wave radars. The authors only discussed three related papers about wireless sensing.

**Questions:**

1. Can the authors provide more discussions about cross-domain sensing?
2. Can the authors compare their methods in terms of detailed efficiency?
3. Can the authors discuss more related works about wireless sensing?

---

> ### Author Response · Authors · 2024-11-23
>
> Thank you for your valuable comments and kind words to our work.
>
> ----
>
> **Q1 The motivation of using Mamba models is not clear enough. The authors mention that Mamba models are more capable of tackling long sequences, while the authors have not discussed the performance of models** **in terms of different lengths of sequences.**
>
> - **Motivation:** Thank you for your comment. The primary motivation for adopting a Mamba-based model lies in its computational efficiency, particularly in terms of memory usage during training, while maintaining high performance.  While attention-based models, such as those leveraging multi-head attention mechanisms, are powerful for capturing long-range dependencies in sequential data, they often come with significant resource demands.
>
>   To illustrate this, we conducted a comparative experiment: we replaced the SSM module in our model with a multi-head attention module while keeping the rest of the architecture unchanged. In our experiments, we found that designing an attention-based model capable of capturing amplitude, phase, and spatial information simultaneously from long sequences of wireless signals exceeded the 24GB memory limit of an NVIDIA RTX 4090 GPU when the batch size was set to 50. This constraint necessitated the reduction of the batch size to 30 for the attention-based model, compared to the Mamba-based model, which operates efficiently at a batch size of 50. Moreover, the Mamba-based model achieves superior performance while maintaining lower computational costs, as demonstrated in the table below:
>
> | **Model**       | **Batch Size** | **GPU Memory Usage** | **MPJPE (mm)** | **Accuracy** | **mAP** |
> | --------------- | -------------- | -------------------- | -------------- | ------------ | ------- |
> | Mamba-based     | 50             | 6.13 GB              | 50.64          | 0.9994       | 0.9991  |
> | Attention-based | 30             | 23.3 GB              | 55.22          | 0.9985       | 0.9972  |
>
> - **Sequence:** Thank you for your insightful suggestion. To evaluate the effect of input sequence length on the performance of our model, we conducted additional experiments with varying sequence lengths.  The results are summarized in the table below:
>
>   |              | 1-frame | 2-frame | 6-frame | 12-frame | 18-frame | 24-frame |
>   | ------------ | ------- | ------- | ------- | -------- | -------- | -------- |
>   | **MPJPE**    | 80.69   | 70.10   | 65.84   | 50.64    | 50.14    | 51.04    |
>   | **Accuracy** | 0.9965  | 0.9974  | 0.9988  | 0.9994   | 0.9995   | 0.9994   |
>   | **mAP**      | 0.9721  | 0.9823  | 0.9968  | 0.9991   | 0.9993   | 0.9992   |
>   | **GFLOPS**   | 0.0382  | 0.0752  | 0.2234  | 0.4458   | 0.6682   | 0.8905   |
>   | **Param(M)** | 1.66    | 1.68    | 1.76    | 1.94     | 2.09     | 2.21     |
>
>   The results indicate that a sequence length of 12 frames provides optimal performance in terms of MPJPE, achieving a good balance between accuracy, computational complexity, and memory usage. Shorter sequences (1-6 frames) lead to reduced performance, while longer sequences (18 frames) show diminishing performance improvements at the cost of significantly increased computational requirements.
>
> ------------------------

---

> > ### Author Response · Authors · 2024-11-23
> >
> > **Q2 A common challenge in RF-based sensing is cross-domain sensing, since RF signals may differ between different environments. I understand that this paper pioneers the use of Mamba models for RF-based sensing, rather than focusing on cross-domain sensing, but the authors should provide more discussions on this.**
> >
> > **(Can the authors provide more discussions about cross-domain sensing?)**
> >
> > To further validate the generalization capability, we collected a comprehensive dataset comprising 48000 pairs of RF frames across different temporal periods, environments, and occlusion conditions (with/without wooden board obstruction). The data includes corresponding optical images, action categories, and human keypoints extracted using AlphaPose. The data was split into a training set and a test set at a 4:1 ratio. The individuals engaged in three complex dynamic activities simultaneously: walking freely, waving hands, and transitioning between standing and sitting positions. This setup provides a more challenging test of the model's capabilities in handling multiple targets with diverse motions.
> >
> > > **Pose Reconstruction Errors (mm) in Three-Person Scenario Without Wooden Board Obstruction**
> >
> > |              | Nose      | Neck      | Shoulder  | Elbow     | Wrist      | Hip       | Knee      | Ankle     | Eye       | Ear       | Average   |
> > | ------------ | --------- | --------- | --------- | --------- | ---------- | --------- | --------- | --------- | --------- | --------- | --------- |
> > | **Person 1** | 78.56     | 79.15     | 81.80     | 111.56    | 170.25     | 79.96     | 81.94     | 82.31     | 77.94     | 79.58     | 93.80     |
> > | **Person 2** | 50.49     | 39.33     | 40.97     | 41.13     | 46.22      | 36.63     | 36.14     | 35.86     | 49.87     | 51.37     | 42.57     |
> > | **Person 3** | 86.70     | 81.95     | 86.24     | 100.87    | 125.49     | 82.98     | 85.81     | 93.32     | 86.20     | 87.35     | 92.51     |
> > | **Average**  | **71.92** | **66.81** | **69.67** | **84.52** | **113.99** | **66.52** | **67.96** | **70.50** | **71.34** | **72.77** | **76.29** |
> >
> > > **Pose Reconstruction Errors (mm) in Three-Person Scenario With Wooden Board Obstruction**
> >
> > |              | Nose      | Neck      | Shoulder  | Elbow     | Wrist      | Hip       | Knee      | Ankle     | Eye       | Ear       | Average   |
> > | ------------ | --------- | --------- | --------- | --------- | ---------- | --------- | --------- | --------- | --------- | --------- | --------- |
> > | **Person 1** | 108.60    | 94.01     | 99.90     | 116.17    | 140.15     | 93.66     | 93.48     | 102.19    | 108.76    | 102.82    | 106.49    |
> > | **Person 2** | 67.06     | 35.24     | 38.23     | 65.89     | 124.37     | 31.97     | 33.23     | 37.06     | 68.14     | 51.54     | 55.73     |
> > | **Person 3** | 96.44     | 94.37     | 94.92     | 96.09     | 99.82      | 90.31     | 81.25     | 78.33     | 98.59     | 99.05     | 92.64     |
> > | **Average**  | **90.70** | **74.54** | **77.68** | **92.72** | **121.45** | **71.98** | **69.32** | **72.53** | **91.83** | **84.47** | **84.95** |
> >
> > Additionally, we provide the qualitative evaluation results under multi-target scenarios in the  **Figure 8 within the Appendix**, which demonstrate that the proposed algorithm can effectively achieve simultaneous reconstruction in multi-target scenarios.
> >
> > -----------

---

> > > ### Author Response · Authors · 2024-11-23
> > >
> > > **Q3 The authors have compared their method with baselines in terms of the number of parameters, while the authors can also compare their detailed efficiency, such as training time, testing time, and throughput.**
> > >
> > > **(Can the authors compare their methods in terms of detailed efficiency?)**
> > >
> > > Thank you for suggesting a more comprehensive efficiency comparison. We have conducted a detailed analysis of efficiency metrics for both multi-stage and single-stage networks:
> > >
> > > Notably, multi-stage networks require a complex data preprocessing phase to transform raw signals into images, making real-time processing challenging. For clarity, we report only the inference time of multi-stage models, excluding data preprocessing time.
> > >
> > > > **Multi-stage Networks**
> > >
> > > | **Model**   | **Training Time (s)** | **Testing Time (s)** | **Throughput (FPS)** | **Params (M)** | **GFLOPS** |
> > > | ----------- | --------------------- | -------------------- | -------------------- | -------------- | ---------- |
> > > | RF-Pose3D   | 3601 × 50 epochs      | 1618                 | 595.60               | 10.91          | 0.9425     |
> > > | ResNet3D-50 | 1402 × 50 epochs      | 334                  | 106.23               | 352.30         | 4.6507     |
> > > | mm-Pose     | 8976 × 50 epochs      | 214                  | 130.95               | 22.07          | 0.3821     |
> > >
> > > > **Single-stage end-to-end networks**
> > >
> > > | **Model**   | **Training Time (s)** | **Testing Time (s)** | **Throughput (FPS)** | **Params (M)** | **GFLOPS** |
> > > | ----------- | --------------------- | -------------------- | -------------------- | -------------- | ---------- |
> > > | RadarFormer | 2041 × 1000 epochs    | 488                  | 44.16                | 12.88          | 0.9299     |
> > > | **RFMamba** | **1967 × 50 epochs**  | **276**              | **60.60**            | **1.94**       | **0.4458** |
> > >
> > > These results clearly demonstrate that RFMamba achieves superior efficiency in terms of training time, testing time, and throughput, while maintaining significantly fewer parameters and lower computational complexity compared to both multi-stage and single-stage baseline models.
> > >
> > > ---------------
> > >
> > > **Q4 Many related works about Wireless Signal Sensing have not been discussed. For example, recent works have involved the use of WiFi signals, ultra-wideband radars, millimeter wave radars. The authors only discussed three related papers about wireless sensing.**
> > >
> > > **(Can the authors discuss more related works about wireless sensing?)**
> > >
> > > Thank you for your valuable feedback. We have revised and expanded the Related Work section to include a broader discussion of recent advancements in wireless sensing, focusing on various RF modalities for human pose estimation. The updated section now includes:
> > >
> > > - **WiFi-based approaches**: GoPose [1], Person-in-WiFi 3D [2], which leverage WiFi signals for non-invasive 3D human pose estimation.
> > > - **Millimeter wave (mmWave) radar systems**: mmPose-NLP [3], HuPR [4], showcasing their capability to estimate skeletal poses using mmWave signals.
> > > - **Ultra-wideband (UWB) radar systems**: UWB-Pose [5], MD-Pose [6], which exploit UWB MIMO radar for efficient through-wall and single-channel pose estimation.
> > >
> > > These additions provide a more comprehensive overview of state-of-the-art methods across RF modalities, highlighting their strengths and contributions to human pose estimation tasks. We hope this expanded discussion sufficiently addresses your concern.
> > >
> > > ---
> > > **References**
> > > [1] Y. Ren, Z. Wang, Y. Wang, S. Tan, Y. Chen, and J. Yang, “GoPose: 3D Human Pose Estimation Using WiFi,” *IMWUT*, 2022.
> > > [2] K. Yan, F. Wang, B. Qian, H. Ding, J. Han, and X. Wei, “Person-in-WiFi 3D: End-to-End Multi-Person 3D Pose Estimation with Wi-Fi,” *CVPR*, 2024.
> > > [3] A. Sengupta and S. Cao, “mmPose-NLP: A Natural Language Processing Approach to Precise Skeletal Pose Estimation Using mmWave Radars,” *TNNLS*, 2023.
> > > [4] S.-P. Lee, N. P. Kini, W.-H. Peng, C.-W. Ma, and J.-N. Hwang, “HuPR: A Benchmark for Human Pose Estimation Using Millimeter Wave Radar,” *WACV*, 2023.
> > > [5] Y. Song, T. Jin, Y. Dai, and X. Zhou, “Efficient Through-Wall Human Pose Reconstruction Using UWB MIMO Radar,” *AWPL*, 2022.  [6] X. Zhou, T. Jin, Y. Dai, Y. Song, and Z. Qiu, “MD-Pose: Human Pose Estimation for Single-Channel UWB Radar,” *TBBIS*, 2023.

---

> > > > ### Comment · Reviewer_oFAp · 2024-11-29
> > > >
> > > > The authors have addressed most of my comments, so I have raised the rating from 5 to 6. However, it would be valuable to validate the proposed method on additional public mmWave datasets or self-collected datasets using wireless devices (e.g., mmWave radar) and to compare it with other state-of-the-art 3D human pose estimation approaches utilizing mmWave radar. Further discussion is needed on the impact of different parameters and deployment environments.

---

> > > > > ### Author Response · Authors · 2024-11-30
> > > > >
> > > > > We sincerely appreciate Reviewer oFAp for the insightful feedback and for raising the score to 6. We have carefully addressed the reviewer’s concerns regarding additional validation and further discussions, as detailed below.
> > > > >
> > > > > As suggested by the reviewer, we evaluate our RFMamba model on mmWave radar signals using the open-source HIBER dataset [1], a public FMCW mmWave radar human perception dataset.  HIBER employs dual mmWave radars with parameters different from our through-wall radar. Specifically, the first radar operates in the range of 77 GHz to 78.23 GHz, and the second in the range of 79 GHz to 80.23 GHz, both with a bandwidth of 1.23 GHz. In contrast, our through-wall radar operates in the range of 0.8 GHz to 2.8 GHz with a bandwidth of 2 GHz. This allowed us to evaluate RFMamba’s generalization capabilities across different radar parameters. The evaluation results, compared with other state-of-the-art mmWave-based methods, are summarized in the following table:
> > > > >
> > > > > | **Model**      | RF-Pose3D | mmPose | RPM   | RFMamba   |
> > > > > | - | - | - | - | - |
> > > > > | **MPJPE (mm)** | 136.8     | 102.6  | 59.20 | **61.44** |
> > > > > | **Params (M)** | 9.492     | 33.08  | 81.67 | **2.29**  |
> > > > >
> > > > > RFMamba achieves an MPJPE of 61.44 mm with only 2.29M parameters, which is comparable to the state-of-the-art RPM model [2] that achieves 59.20 mm MPJPE but requires a larger 81.67M parameters. Despite the substantial differences in signal type (FMCW vs. SFCW), operating frequency (77-80 GHz vs. 0.8-2.8 GHz), and bandwidth (1.23 GHz vs. 2 GHz), RFMamba maintains comparable performance across both systems. These results validate the model's robust generalization across **different radar parameters**, as well as its potential applicability to other wireless devices.
> > > > >
> > > > > We further conduct the comparison under the non-occluded deployment environment in self-collected THP-F (free-space) dataset. The results for pose estimation performance in non-occluded environment are summarized below:
> > > > >
> > > > > | Method      | Nose      | Neck      | Shoulder  | Elbow     | Wrist     | Hip       | Knee      | Ankle     | Eye       | Ear       | Mean      |
> > > > > | - | - | - | - | - | - | - | - | - | - | - | - |
> > > > > | RF-Pose3D   | 75.96     | 65.89     | 71.70     | 91.98     | 132.56    | 66.05     | 68.81     | 83.34     | 78.14     | 72.33     | 81.76     |
> > > > > | mm-Pose     | 115.57    | 100.09    | 126.48    | 166.45    | 206.34    | 109.53    | 111.29    | 114.25    | 115.85    | 108.76    | 129.64    |
> > > > > | ResNet3D-50 | 136.96    | 109.70    | 128.53    | 127.47    | 159.77    | 113.47    | 129.56    | 131.46    | 132.44    | 131.76    | 130.86    |
> > > > > | RadarFormer | 213.33    | 202.38    | 205.40    | 221.46    | 262.95    | 202.95    | 196.39    | 197.58    | 214.63    | 207.28    | 212.94    |
> > > > > | **RFMamba** | **63.94** | **54.61** | **56.37** | **63.53** | **83.44** | **53.81** | **50.34** | **56.02** | **63.84** | **59.60** | **60.69** |
> > > > >
> > > > > In the **non-occluded deployment environment**, RFMamba consistently outperforms baseline methods by achieving lower reconstruction errors across all keypoints, highlighting its superior capability to accurately capture human poses.
> > > > >
> > > > > To further validate the generalization across different deployment environments, we collected a comprehensive dataset comprising 48000 pairs of RF frames across different environments and occlusion conditions (with/without wooden board obstruction). The data includes corresponding optical images, action categories, and human keypoints extracted using AlphaPose. The data was split into a training set and a test set at a 4:1 ratio. The individuals engaged in three complex dynamic activities: walking freely, waving hands, and transitioning between standing and sitting positions. This setup provides a more challenging test of the model's capabilities in handling new deployment environments. The results are summarized below:
> > > > >
> > > > > |                         | Nose      | Neck      | Shoulder  | Elbow     | Wrist      | Hip       | Knee      | Ankle     | Eye       | Ear       | Average   |
> > > > > | - | - | - | - | - | - | - | - | - | - | - | - |
> > > > > | **Without Obstruction** | 71.92 | 66.81| 69.67 | 84.52 | 113.99 | 66.52 | 67.96 | 70.50 | 71.34 | 72.77 | 76.29 |
> > > > > | **With Obstruction**    | 90.70 | 74.54 | 77.68 | 92.72 | 121.45 | 71.98 | 69.32 | 72.53 | 91.83 | 84.47 | 84.95 |
> > > > >
> > > > > Additionally, we present the qualitative evaluation results in **Figure 8 within the Appendix**, which illustrate that the proposed algorithm can effectively achieve human perception in the **new deployment environments**.
> > > > >
> > > > > ---
> > > > >
> > > > > **References**
> > > > > [1] Wu, Z., Zhang, D., Xie, C., Yu, C., Chen, J., Hu, Y., and Chen, Y., "RFMask: A simple baseline for human silhouette segmentation with radio signals," *IEEE Transactions on Multimedia*, vol. 25, pp. 4730–4741, 2022.
> > > > >
> > > > > [2] Xie, C., Zhang, D., Wu, Z., Yu, C., Hu, Y., and Chen, Y., "RPM: RF-Based Pose Machines," *IEEE Transactions on Multimedia*, vol. 26, pp. 637–649, 2024, doi: 10.1109/TMM.2023.3268376.

---

### Official Review · Reviewer_oFAp · 2024-11-02

**Soundness:** 3
**Presentation:** 2
**Contribution:** 2
**Rating:** 6
**Confidence:** 4

**Summary:**

This paper presents a state space model designed for RF-based human perception. The authors introduce various modules, incorporating both frequency domain and spatio-temporal domain components. Additionally, they develop a six-way scanning method within the frequency modeling branch. Experimental results demonstrate the effectiveness of the proposed approach.

**Strengths:**

(1) This paper is the first to apply the Mamba method for RF-based sensing, marking a novel contribution to the field.

(2) The authors provide comprehensive discussions for each module in the proposed framework.

(3) A six-way scanning strategy in the frequency modeling branch is proposed, enabling adaptive selection of the most informative frequency cues.

(4) Real-world experimental results demonstrate the effectiveness of the proposed method in terms of both accuracy and parameter efficiency.

**Weaknesses:**

(1) The introduction sections on RF sensing and SFCW are underdeveloped. A more detailed discussion on the model and data specifics for SFCW radar is needed.

(2) The related work section is limited. The authors should include a review of state-of-the-art wireless signal sensing work relevant to RF pose estimation and clarify distinctions between the proposed model and other Mamba-based approaches.

(3) The framework of the model requires a more detailed discussion, and the training algorithm should be included in the paper.

(4) The experimental results could benefit from further enhancement.

**Questions:**

(1)The authors primarily use a custom MIMO (SFCW) radar to validate their proposed method. Consequently, the paper’s title, which references RF-based human-centric perception, may overstate the scope, as other radar types and RF-based sensing modalities (e.g., Wi-Fi, RFID, LoRa) are not examined. The authors should clarify their motivation for selecting SFCW radar, including its advantages over existing options like TI mmWave radar.

(2) The introduction of SFCW radar in Section 3.1.1 lacks clarity. For example, it is unclear how Eq. 3 represents the signal across three key dimensions: fast time (distance), antenna (azimuth and elevation), and slow time (velocity across echoes). A more detailed explanation in this section would improve understanding.

(3) In the related work section, more state-of-the-art studies on wireless signal sensing relevant to RF pose estimation should be reviewed. The authors should also discuss distinctions between the proposed Mamba method and other Mamba-based methods.

(4) For the framework depicted in Fig. 2, while each block is briefly explained, more specific functionality should be provided, including details on the SiLU block. Additionally, the loss function and training approach require elaboration. For instance, the rationale for using different task losses to define the joint training loss in Eq. 11 should be clarified. Since the code is not provided, a detailed description of the training algorithm is needed.

(5) Though the proposed method is tested with a real-world setup, it would benefit from validation using a public dataset, such as an mmWave radar dataset. The authors should also consider benchmarking against advanced transformer models and other Mamba methods. Furthermore, the evaluation on generalization and multi-person scenarios is limited; additional tests across different time (different days) and involving more than three individuals would strengthen the findings.

---

> ### Author Response · Authors · 2024-11-23
>
> Thank you for your valuable feedback on our paper.
>
> ---
>
> **Q1** **The introduction sections on RF sensing and SFCW are underdeveloped. A more detailed discussion on the model and data specifics for SFCW radar is needed.**
>
> As suggested by the reviewer, we have included more discussion on RF sensing and SFCW radar in the introduction section.  Additionally, we would like to highlight that our proposed framework is not restricted to SFCW radar; it is also applicable to FMCW radar. For further details, please refer to our response to Q7, where we present additional experiments conducted on an FMCW radar dataset.
>
> --------------------
>
> **Q2 The related work section is limited. The authors should include a review of state-of-the-art wireless signal sensing work relevant to RF pose estimation and clarify distinctions between the proposed model and other Mamba-based approaches.**
>
> **(In the related work section, more state-of-the-art studies on wireless signal sensing relevant to RF pose estimation should be reviewed. The authors should also discuss distinctions between the proposed Mamba method and other Mamba-based methods.)**
>
> Thank you for your comments. As suggested by the reviewer, we have thoroughly revised the **Related Work** section to include a detailed discussion of state-of-the-art wireless sensing methods for pose estimation [1-6]. Additionally, we have clarified the distinctions between RFMamba and other Mamba-based methods. Specifically, unlike existing Mamba-based models, RFMamba addresses the unique challenges of RF signals through novel components like the RF-SSM block and omni-dimensional scanning strategy. This approach enables efficient processing of both amplitude and phase information in long RF sequences, a crucial aspect not addressed by Mamba models in other domains.

---

> ### Author Response · Authors · 2024-11-23
>
> **Q3 The framework of the model requires a more detailed discussion, and the training algorithm should be included in the paper.**
>
> **(For the framework depicted in Fig. 2, while each block is briefly explained, more specific functionality should be provided, including details on the SiLU block. Additionally, the loss function and training approach require elaboration. For instance, the rationale for using different task losses to define the joint training loss in Eq. 11 should be clarified. Since the code is not provided, a detailed description of the training algorithm is needed.)**
>
> Thank you for your valuable feedback. We have revised the manuscript to include a detailed discussion of the model framework, with additional explanations of the components depicted in Fig. 2. Furthermore, we have elaborated on the loss function and training approach to provide a more comprehensive and clear presentation of our methodology, as detailed below:
>
> - **Framework:**
>
>   We have improved Section 3.2 by adding a more detailed description of the model framework.
>
> - **The SiLU:**
>   The Sigmoid Linear Unit (SiLU) functions as an activation branch that processes the feature representations. Its output is then multiplicatively combined with the spatiotemporal features captured by the ST Scan module, enabling adaptive feature modulation through the non-linear activation characteristics of SiLU.
>
> - **Loss Function:**
>   The joint training loss integrates task-specific objectives where pose estimation loss (MPJPE) measures the Euclidean distance between predicted and ground truth joint positions, while action recognition and ReID losses utilize cross-entropy for classification tasks. This multi-task formulation enables simultaneous optimization of spatial accuracy (pose), temporal dynamics (action), and identity preservation (ReID) within a unified learning framework.
>
> - **Training Approach:**
>   RFMamba is implemented with PyTorch and trained using an NVIDIA RTX4090 GPU. We train RFMamba on our collected dataset for 50 epochs using the Adam optimizer, with a batch size of 50. The initial learning rate is established at \(2 \times 10^{-3}\), and decays by a factor of 0.5 (gamma) every 10 epochs using StepLR scheduler. We stack 12 consecutive frames (a 1-second time window) as input and split the dataset into training and test sets with a ratio of 4:1.
>
> ------------
>
> **Q4 The experimental results could benefit from further enhancement.**
>
> As suggested by the reviewer, we have conducted additional experiments to further strengthen our findings. These include:
>
> 1. Evaluations on a public mmWave radar dataset.
> 2. Benchmarking against advanced transformer models and other Mamba methods.
> 3. Assessments of generalization across different timeframes and multi-person scenarios.
>
> For detailed experimental results, please refer to **Q7**.
>
> --------------------
>
> **Q5 The authors primarily use a custom MIMO (SFCW) radar to validate their proposed method. Consequently, the paper’s title, which references RF-based human-centric perception, may overstate the scope, as other radar types and RF-based sensing modalities (e.g., Wi-Fi, RFID, LoRa) are not examined. The authors should clarify their motivation for selecting SFCW radar, including its advantages over existing options like TI mmWave radar.**
>
> We agree with the reviewer that existing studies have utilized various RF signals for human sensing, including Stepped-Frequency Continuous Wave (SFCW) radar, Frequency-Modulated Continuous Wave (FMCW) radar, WiFi, and LoRa signals. However, among these, SFCW and FMCW signals are particularly well-suited for fine-grained human sensing because their larger bandwidth enables higher range resolution. Additionally, we would like to highlight that our proposed framework is not restricted to SFCW radar; it is also applicable to FMCW radar.
>
> We have chosen SFCW radar operating at relatively low frequencies (0.8-2.8 GHz) due to its superior penetration capabilities, which enable both through-wall and non-occluded sensing applications. In contrast, TI mmWave radars operate at much higher frequencies (e.g., 77 GHz), where penetration ability is significantly limited, making them primarily suitable for non-occluded scenarios.
>
> ---------------

---

> ### Author Response · Authors · 2024-11-23
>
> **Q6 The introduction of SFCW radar in Section 3.1.1 lacks clarity. For example, it is unclear how Eq. 3 represents the signal across three key dimensions: fast time (distance), antenna (azimuth and elevation), and slow time (velocity across echoes). A more detailed explanation in this section would improve understanding.**
>
> We have revised Section 3.1.1 to clarify how the inverse Fourier transform maps the signal to different physical dimensions. Specifically, we now explicitly state that the transformation converts the frequency-spatial domain signal (X(u,v)) to the range-angle domain (S(h,w)), where h corresponds to the angle dimension from antenna array spatial sampling, w represents the range dimension from frequency stepping, and velocity information is derived from tracking range variations across consecutive frames.
>
> --------------
>
> **Q7 Though the proposed method is tested with a real-world setup, it would benefit from validation using a public dataset, such as an mmWave radar dataset. The authors should also consider benchmarking against advanced transformer models and other Mamba methods. Furthermore, the evaluation on generalization and multi-person scenarios is limited; additional tests across different time (different days) and involving more than three individuals would strengthen the findings.**
>
> Thank you for your thoughtful comment. As suggested by the reviewer, we have evaluated our RFMamba model on FMCW mmWave radar signals using the open-source HIBER dataset [1]. Additionally, we have conducted an empirical comparison of our model with the SSM-based VMamba, the Transformer-based ViT, and the Transformer-based RadarFormer. Furthermore, we have validated the generalization capability across different time (different days) involving more than three individuals, as detailed below:
>
> - **HIBER Dataset:**
>   HIBER is a public FMCW mmWave radar human perception dataset that includes a diverse range of environments, users, occlusions, and actions. We evaluated RFMamba on this dataset and compared with state-of-the-art FMCW mmWave-based methods.The evaluation results are summarized  in the following table. RFMamba achieves a performance of 61.44 mm MPJPE while utilizing only 2.29M parameters and 1.48 GFLOPs. This is comparable to the state-of-the-art model RPM [2], which achieves 59.20 mm MPJPE but requires 81.67M parameters and 2148.44 GFLOPs. These findings underscore RFMamba's strong generalization capabilities across various signal types and domains.
>
>   | **Model**   | **MPJPE (mm)** | **Params (M)** | **GFLOPS** |
>   | ----------- | -------------- | -------------- | ---------- |
>   | RF-Pose3D   | 136.8          | 9.492          | 39.58      |
>   | mmPose      | 102.6          | 33.08          | 0.38       |
>   | RPM         | 59.20          | 81.67          | 2148.44    |
>   | **RFMamba** | **61.44**      | **2.29**       | **1.48**   |
>
> - **Advanced Models:**
>
>   Thank you for your insightful comments, we have conducted a empirical comparison of our model with the SSM-based **VMamba** [3], the Transformer-based **ViT**[4], and the Transformer-based **RadarFormer**[5]. The detailed computational costs are presented in the table below.  As shown in the table, compared to other similar models, our RFMamba achieves competitive performance with an MPJPE of 50.64 mm, an accuracy of 0.9994, and an mAP of 0.9991, while utilizing fewer parameters (1.94M), requiring only 0.4458 GFLOPs.
>
> | **Model**    | **MPJPE**  **(mm)** | **Accuracy** | **mAP**    | **Params  (M)** | **GFLOPS** |
> | ------------ | ------------------- | ------------ | ---------- | --------------- | ---------- |
> | RardarFormer | 258.90              | 0.8918       | 0.4817     | 12.88           | 0.9299     |
> | ViT          | 64.36               | 0.9930       | 0.9907     | 110.69          | 22.6635    |
> | VMamba       | 61.74               | 0.9973       | **0.9993** | 4.82            | 0.5180     |
> | **RFMamba**  | **50.64**           | **0.9994**   | 0.9991     | **1.94**        | **0.4458** |

---

> > ### Author Response · Authors · 2024-11-23
> >
> > - **Generalization Capability:**
> >
> >   To further validate the generalization capability, we collected a comprehensive dataset comprising 48000 pairs of RF frames across different temporal periods, environments, and occlusion conditions (with/without wooden board obstruction). The data includes corresponding optical images, action categories, and human keypoints extracted using AlphaPose. The data was split into a training set and a test set at a 4:1 ratio. The individuals engaged in three complex dynamic activities simultaneously: walking freely, waving hands, and transitioning between standing and sitting positions. This setup provides a more challenging test of the model's capabilities in handling multiple targets with diverse motions.
> >
> >   > **Pose Reconstruction Errors (mm) in Three-Person Scenario Without Wooden Board Obstruction**
> >
> >   |              | Nose      | Neck      | Shoulder  | Elbow     | Wrist      | Hip       | Knee      | Ankle     | Eye       | Ear       | Average   |
> >   | ------------ | --------- | --------- | --------- | --------- | ---------- | --------- | --------- | --------- | --------- | --------- | --------- |
> >   | **Person 1** | 78.56     | 79.15     | 81.80     | 111.56    | 170.25     | 79.96     | 81.94     | 82.31     | 77.94     | 79.58     | 93.80     |
> >   | **Person 2** | 50.49     | 39.33     | 40.97     | 41.13     | 46.22      | 36.63     | 36.14     | 35.86     | 49.87     | 51.37     | 42.57     |
> >   | **Person 3** | 86.70     | 81.95     | 86.24     | 100.87    | 125.49     | 82.98     | 85.81     | 93.32     | 86.20     | 87.35     | 92.51     |
> >   | **Average**  | **71.92** | **66.81** | **69.67** | **84.52** | **113.99** | **66.52** | **67.96** | **70.50** | **71.34** | **72.77** | **76.29** |
> >
> >   > **Pose Reconstruction Errors (mm) in Three-Person Scenario With Wooden Board Obstruction**
> >
> >   |              | Nose      | Neck      | Shoulder  | Elbow     | Wrist      | Hip       | Knee      | Ankle     | Eye       | Ear       | Average   |
> >   | ------------ | --------- | --------- | --------- | --------- | ---------- | --------- | --------- | --------- | --------- | --------- | --------- |
> >   | **Person 1** | 108.60    | 94.01     | 99.90     | 116.17    | 140.15     | 93.66     | 93.48     | 102.19    | 108.76    | 102.82    | 106.49    |
> >   | **Person 2** | 67.06     | 35.24     | 38.23     | 65.89     | 124.37     | 31.97     | 33.23     | 37.06     | 68.14     | 51.54     | 55.73     |
> >   | **Person 3** | 96.44     | 94.37     | 94.92     | 96.09     | 99.82      | 90.31     | 81.25     | 78.33     | 98.59     | 99.05     | 92.64     |
> >   | **Average**  | **90.70** | **74.54** | **77.68** | **92.72** | **121.45** | **71.98** | **69.32** | **72.53** | **91.83** | **84.47** | **84.95** |
> >
> >   Additionally, we provide the qualitative evaluation results under multi-target scenarios in the  **Figure 8 within the Appendix**, which demonstrate that the proposed algorithm can effectively achieve simultaneous reconstruction in multi-target scenarios.
> >
> > For scenarios involving more than three people, we have decided to leave this aspect for future work. This is because the multi-camera system used to collect pose ground truth becomes unstable in such cases, compromising the quality of the ground truth data.
> >
> > ------------------------
> >
> > **References**
> > [1] Wu, Z., Zhang, D., Xie, C., Yu, C., Chen, J., Hu, Y. and Chen, Y., 2022. RFMask: A simple baseline for human silhouette segmentation with radio signals. *IEEE Transactions on Multimedia*, 25, pp.4730-4741.
> > [2] C. Xie, D. Zhang, Z. Wu, C. Yu, Y. Hu and Y. Chen, "RPM: RF-Based Pose Machines," in *IEEE Transactions on Multimedia*, vol. 26, pp. 637-649, 2024, doi: 10.1109/TMM.2023.3268376.
> > [3] Y. Liu, Y. Tian, Y. Zhao, H. Yu, L. Xie, Y. Wang, Q. Ye, and Y. Liu, "VMamba: Visual State Space Model," *arXiv preprint arXiv:2401.10166*, 2024.
> > [4] A. Dosovitskiy, L. Beyer, A. Kolesnikov, D. Weissenborn, X. Zhai, T. Unterthiner, M. Dehghani, M. Minderer, G. Heigold, S. Gelly, J. Uszkoreit, and N. Houlsby, "An image is worth 16x16 words: Transformers for image recognition at scale," *arXiv preprint arXiv:2010.11929*, 2020.
> > [5] Z. Zheng, D. Zhang, X. Liang, X. Liu and G. Fang, "RadarFormer: End-to-End Human Perception With Through-Wall Radar and Transformers," in *IEEE Transactions on Neural Networks and Learning Systems*, doi: 10.1109/TNNLS.2023.3314031.

---

### Official Review · Reviewer_uCrw · 2024-11-03

**Soundness:** 3
**Presentation:** 2
**Contribution:** 3
**Rating:** 6
**Confidence:** 3

**Summary:**

The paper proposes RFMamba, a frequency-aware state space model (SSM) for human-centric perception using RF signals. RFMamba addresses the challenges of processing long-sequence RF data, with an innovative dual-branch SSM block that handles frequency and spatio-temporal cues for better accuracy in human tracking tasks. The model is evaluated on downstream tasks and shows significant performance improvements over existing methods. The proposed method is inspiring and paper is well-structured, but could improve in clarity and visual representation.

**Strengths:**

The model offers a practical solution to handle long-sequence RF signals with linear complexity. The dual-branch design of the block integrates both frequency and spatio-temporal information, addressing the core challenge for RF signal representation learning. The proposed RFMamba shows clear improvements in localization, recognition, and action detection tasks.

**Weaknesses:**

1) Some equations and model parameters are not sufficiently explained. This makes it a bit hard to grasp the model's operation and how it ties into the broader methodology.
For example, some suggestions for modification:
- For Figure 1, it would be better to enlarge the font sizes of labels and titles.
- In Section 3.1.1, after the introduction of frequency and spatial analysis in RF sensing, would be better to connect more to the proposed method and methodology.
- In Section 3.1.2, eq (4), explain better, not clear what are h and h' meaning, and model parameters A, B, C, D's usage, also can connect more to the overall methodology.

2) While the paper claims linear complexity, it's a bit unclear the resulting computational costs compared with similar models. Some theoretical or empirical comparison would help better clarify its practical deployability.

3) The model's applicability to other domains and signal types can make a straightforward discussion to complete the methodology and usage range.

**Questions:**

1) Can the model specifically deal with extreme RF environments in practical use? such as high interference or low signal-to-noise ratios? and what about the demanded resources (data and computation) if adapted to practical scenarios?

2) How does RFMamba's computational efficiency compare to similar SSM or Transformer-based models in terms of real-time performance?

3) How generalizable is RFMamba to other RF-based applications beyond human perception?

---

> ### Author Response · Authors · 2024-11-23
>
> We sincerely thank you for the valuable comments and suggestions.
>
> ---
>
> **Q1 Some equations and model parameters are not sufficiently explained. This makes it a bit hard to grasp the model's operation and how it ties into the broader methodology. For example, some suggestions for modification:**
>
> - **For Figure 1, it would be better to enlarge the font sizes of labels and titles.**
>
> - **In Section 3.1.1, after the introduction of frequency and spatial analysis in RF sensing, would be better to connect more to the proposed method and methodology.**
>
> - **In Section 3.1.2, eq (4), explain better, not clear what are h and h' meaning, and model parameters A, B, C, D's usage,** **also can connect more to the overall methodology.**
>
> We thank the reviewer for the suggestions and have revised the paper accordingly.
>
> - We have enlarged the font sizes of labels and titles of Figure 1.
>
> - We have revised Section 3.1.1 to better connect the frequency and spatial analysis with our proposed method. We now explicitly mention how RFMamba leverages the multi-dimensional information extracted from RF signals, highlighting the role of our RF-SSM block and omni-dimensional scanning strategy in comprehensively modeling human motion.
>
> - We have revised Section 3.1.2 to provide a clearer explanation:
>   - **h(t):** The hidden state that accumulates historical information from the input RF signal sequence.
>   - **h'(t):** Describes how this hidden state evolves over time.
>   - **Parameters {A, B, C, D}:**
>     - **A:** Controls how the hidden state transitions between time steps.
>     - **B:** Maps input signals to state space updates.
>     - **C:** Projects the hidden state to generate outputs.
>     - **D:** Provides skip connections from input to output.
>
> This formulation, inspired by Mamba's selective state space modeling, is particularly effective for RF signals as it can: 1) Process long sequences with linear complexity; 2) Capture both frequency-domain and spatial-temporal patterns; 3) Adapt to varying signal characteristics through learned parameters.
>
> ---
>
> **Q2 While the paper claims linear complexity, it's a bit unclear the resulting computational costs compared with similar models. Some theoretical or empirical comparison would help better clarify its practical deployability.**
>
> **(How does RFMamba's computational efficiency compare to similar SSM or Transformer-based models in terms of real-time performance?)**
>
> As suggested by the reviewer, we have conducted a empirical comparison of our model with the SSM-based **VMamba** [1], the Transformer-based **ViT**[2], and the Transformer-based **RadarFormer**[3]. The detailed computational costs are presented in the table below.
>
> As shown in the following table, compared to other similar models, our RFMamba achieves competitive performance with an MPJPE of 50.64 mm, an accuracy of 0.9994, and an mAP of 0.9991, while utilizing fewer parameters (1.94M), requiring only 0.4458 GFLOPs.
>
> | **Model**    | **MPJPE**  **(mm)** | **Accuracy** | **mAP**    | **Params  (M)** | **GFLOPS** |
> | ------------ | ------------------- | ------------ | ---------- | --------------- | ---------- |
> | RardarFormer | 258.90              | 0.8918       | 0.4817     | 12.88           | 0.9299     |
> | ViT          | 64.36               | 0.9930       | 0.9907     | 110.69          | 22.6635    |
> | VMamba       | 61.74               | 0.9973       | **0.9993** | 4.82            | 0.5180     |
> | **RFMamba**  | **50.64**           | **0.9994**   | 0.9991     | **1.94**        | **0.4458** |
>
> ---
>
> **Q3 The model's applicability to other domains and signal types can make a straightforward discussion to complete the methodology and usage range.**
>
> As suggested by the reviewer, we have evaluated our RFMamba model on FMCW mmWave radar signals using the open-source HIBER dataset [1].
>
> HIBER is a publicly available mmWave human perception dataset that includes a diverse range of environments, users, occlusions, and actions. The evaluation results are summarized in the table.
>
> RFMamba achieves a performance of 61.44 mm MPJPE while utilizing only 2.29M parameters and 1.48 GFLOPs. This is comparable to the state-of-the-art model RPM[4], which achieves 59.20 mm MPJPE but requires 81.67M parameters and 2148.44 GFLOPs. These findings underscore RFMamba's strong generalization capabilities across various signal types and domains.
>
> | Model       | MPJPE (mm) | Params (M) | GFLOPs   |
> | ----------- | ---------- | ---------- | -------- |
> | RF-Pose3D   | 136.8      | 9.492      | 39.58    |
> | mmPose      | 102.6      | 33.08      | 0.38     |
> | RPM         | 59.20      | 81.67      | 2148.44  |
> | **RFMamba** | **61.44**  | **2.29**   | **1.48** |
>
> ---

---

> > ### Author Response · Authors · 2024-11-23
> >
> > **Q4 Can the model specifically deal with extreme RF environments in practical use? such as high interference or low signal-to-noise ratios? and what about the demanded resources (data and computation) if adapted to practical scenarios?**
> >
> > To evaluate the model’s robustness under extreme RF conditions, we simulated real-world interference by adding Gaussian white noise to the raw RF signals and tested the model’s performance at varying signal-to-noise ratios (SNRs): 0dB, 5dB, and 10dB.
> >
> > The results are summarized in the table below, indicate that while higher noise levels lead to some performance degradation, RFMamba consistently achieves superior performance. Even in highly challenging scenarios with 0dB SNR, the model demonstrates competitive results across all tasks, highlighting its potential for real-world deployment in demanding RF environments.
> >
> > | **SNR**  | **0dB** | **5dB** | **10dB** |
> > | -------- | ------- | ------- | -------- |
> > | MPJPE    | 92.92   | 70.50   | 64.62    |
> > | Accuracy | 0.9958  | 0.9981  | 0.9982   |
> > | mAP      | 0.9748  | 0.9842  | 0.9922   |
> >
> > ---
> >
> > **Q5 How generalizable is RFMamba to other RF-based applications beyond human perception?**
> >
> > As a general-purpose RF signal encoder, RFMamba can directly process raw RF data, thus supporting flexible RF signal types. Moreover, since the prediction head can be easily changed to adapt to different applications, we believe it has the potential to be applicable to a wide range of RF-based tasks.
> >
> > ---
> >
> > **References**
> >
> > [1] Y. Liu, Y. Tian, Y. Zhao, H. Yu, L. Xie, Y. Wang, Q. Ye, and Y. Liu, "VMamba: Visual State Space Model," *arXiv preprint arXiv:2401.10166*, 2024.
> >
> > [2] A. Dosovitskiy, L. Beyer, A. Kolesnikov, D. Weissenborn, X. Zhai, T. Unterthiner, M. Dehghani, M. Minderer, G. Heigold, S. Gelly, J. Uszkoreit, and N. Houlsby, "An image is worth 16x16 words: Transformers for image recognition at scale," *arXiv preprint arXiv:2010.11929*, 2020.
> >
> > [3] Z. Zheng, D. Zhang, X. Liang, X. Liu and G. Fang, "RadarFormer: End-to-End Human Perception With Through-Wall Radar and Transformers," in *IEEE Transactions on Neural Networks and Learning Systems*, doi: 10.1109/TNNLS.2023.3314031.
> >
> > [4] C. Xie, D. Zhang, Z. Wu, C. Yu, Y. Hu and Y. Chen, "RPM: RF-Based Pose Machines," in *IEEE Transactions on Multimedia*, vol. 26, pp. 637-649, 2024, doi: 10.1109/TMM.2023.3268376.

---

### Author Response · Authors · 2024-11-24

Author Rebuttal

We sincerely thank all reviewers for their valuable comments and suggestions. We appreciate that reviewers recognized our work as "novel", "well-structured", and "marking a novel contribution to the field" with "comprehensive discussions". Each reviewer's feedback has been carefully addressed individually. The manuscript has been revised according to the suggestions provided. Here is a summary of our main revisions:

1. Model Analysis & Technical Details:

- Added detailed sequence length analysis showing optimal performance at 12 frames (MPJPE: 50.64mm)
- Clarified framework design choices and parameter meanings (h(t), h'(t), A, B, C, D)
- Provided comprehensive efficiency metrics comparing with baselines:
  * RF-adaptive SSM-based RFMamba : 50.64 mm MPJPE, 1.94M params, 0.4458 GFLOPs
  * Similar SSM-based VMamba : 61.74 mm MPJPE, 4.82M params, 0.5180 GFLOPs
  * Transformer-based ViT : 64.36 mm MPJPE, 110.69M params, 22.6635 GFLOPs

2. Generalization & Cross-Domain Capability:

- Evaluated on public HIBER FMCW dataset:
  * RFMamba: 61.44 mm MPJPE, 2.29M params
  * SOTA RPM: 59.20 mm MPJPE but requires 81.67M params
- Validated multi-person scenarios with/without occlusion
- Enhanced baseline comparisons across different scenarios
- Demonstrated adaptation to new environments

3. Dataset & Experimental Details:

- Clarified data composition: 19 activities (3 dynamic, 16 static)
- Added IRB approval and informed consent information
- Detailed training protocol and evaluation metrics
- Plan to release THP dataset within 3 months

The revisions demonstrate RFMamba's advantages in efficiency, accuracy, and generalization while addressing all major concerns. We look forward to further discussions and are happy to address any additional questions.

Best regards,
Authors

---

### Meta-Review · Area_Chair_YhrP · 2024-12-08

**Metareview:**

The paper presents RFMamba, a novel frequency-aware state space model for RF-based human-centric perception (HCP), designed to process radio frequency (RF) signals for tasks like human pose estimation, activity recognition, and person re-identification (ReID). Leveraging the capabilities of state space models (SSMs), RFMamba uses a dual-branch architecture that combines frequency and spatiotemporal modeling to effectively capture the characteristics of RF signals. The model introduces an omni-dimensional scanning mechanism, selectively focusing on informative frequency cues. Experiments on the newly introduced THP dataset (covering both free-space and wall-occlusion scenarios) demonstrate RFMamba’s superior performance over existing methods in accuracy. After revision, I do not see major flaws in this paper. The method shows the effectiveness on downstream tasks and the experiments are sufficient.

**Additional Comments On Reviewer Discussion:**

**Reviewer uCrw:**
1. More clarity needed on key mathematical details and model components.
2. Better explanation linking the theoretical model foundation to the overall methodology.

**Reviewer oFAp:**
1. Clarify the motivation for using SFCW radar and address the title’s scope given that other RF modalities are not tested.
2. Enhance explanation of SFCW radar signal representation across time, antenna, and velocity dimensions.
3. Review more state-of-the-art wireless sensing studies, clarify distinctions among Mamba-based methods, and provide more detail on framework components, loss function, training details, and public dataset validations.

**Reviewer 5Q2S:**
1. Clarify the motivation for using Mamba models and discuss performance across different sequence lengths.
2. Provide more discussion on cross-domain challenges and differences in RF signals across environments.

**Reviewer aiy3:**
1. Analyze the effect of sequence length on performance, efficiency, and model stability.
2. Investigate why RadarFormer underperforms and whether Transformer-based models could be improved with additional adaptations.
3. Provide more detail on activity distributions, environmental diversity, dataset release plans, real-time performance, and direct generalization comparisons with baselines.

The authors revise the manuscript and provide the rebuttal:
1. **Model Clarity & Technical Details:**
   - **Reviewers:** Requested more explanation of model parameters and rationale, as well as analysis of the effect of sequence length.
   - **Authors:** Provided detailed parameter definitions, clarified design choices, and showed that 12-frame sequences yield optimal accuracy (MPJPE: 50.64 mm). Efficiency metrics were also added.

2. **Generalization & Cross-Domain Capability:**
   - **Reviewers:** Questioned how well the model would generalize to new datasets and scenarios, including multi-person and occluded setups.
   - **Authors:** Tested on a public FMCW dataset and under multi-person/occlusion conditions. While slightly less accurate than one SOTA method, RFMamba used far fewer parameters, showing strong adaptability and efficiency.

3. **Dataset & Experimental Details:**
   - **Reviewers:** Asked for clarity on dataset composition, IRB approvals, training protocols, and plans for data release.
   - **Authors:** Provided a detailed breakdown of activities, confirmed IRB/informed consent, and committed to releasing the THP dataset within 3 months.

**Final Decision Influence:**
All reviewer concerns were addressed with clear explanations, new experiments, and additional documentation. I think the paper can be accepted in the current format.

---

### Decision · Program_Chairs · 2025-01-22

Accept (Poster)